# Associations Among Developmental Coordination Disorder Traits, Neurodevelopmental Difficulties and University Personality Inventory Scores in Undergraduate Students at a Japanese National University: A Cross-Sectional Correlational Study

**DOI:** 10.3390/brainsci15080895

**Published:** 2025-08-21

**Authors:** Masanori Yasunaga, Ryutaro Higuchi, Keita Kusunoki, Naoto Mochizuki

**Affiliations:** 1Health and Counseling Center, The University of Osaka, Toyonaka 560-0043, Japan; 2Student Support Center, Bukkyo University, Kyoto 603-8301, Japan

**Keywords:** developmental coordination disorder, attention deficit hyperactivity disorder, higher education institutions, neurodevelopmental disorders, trait screening

## Abstract

Background/Objectives: Developmental coordination disorder (DCD) commonly co-occurs with attention deficit hyperactivity disorder (ADHD), autism spectrum disorder (ASD), and specific learning disorder (SLD), affecting academics, mental health, and lifestyle. Although screeners such as the Adolescents and Adults Coordination Questionnaire (AAC-Q) are widely used elsewhere, recognition and support for DCD in Japan remain limited. This study examined associations among DCD traits, ADHD/ASD-related difficulties, learning difficulties, mental health difficulties, and lifestyle factors in university students. Methods: We conducted a web-based survey of 16,295 students; responses from 527 were analyzed. The instruments used for the analysis included the AAC-Q, short forms of ADHD and ASD Difficulty Scales, the 7-item Learning Difficulty Scale for Postsecondary Students and 10-item Scale for Childhood Learning Difficulties, and the University Personality Inventory (UPI). Nonparametric and Spearman’s rank correlations were performed. Results: DCD traits were observed at 7.4% (AAC-Q ≥ 32). This is a screen-positive proportion only. AAC-Q scores correlated strongly with ADHD difficulties (*r* = 0.65), moderately to strongly with ASD difficulties (*r* = 0.55), and moderately with mental health difficulties measured by the UPI (*r* = 0.41). Conclusions: These findings suggest that identifying DCD traits at university entry may be associated with greater student self-understanding and improved access to appropriate support. They support targeted DCD screening alongside ADHD/ASD screening at university entry, rather than universal screening.

## 1. Introduction

Individuals with developmental coordination disorder (DCD) fail to acquire sufficiently skilled motor abilities, and this shortfall adversely affects their daily lives and academic performance. It cannot be explained by low disability or general medical conditions [1]. The estimated prevalence of developmental coordination disorder (DCD) is 2–6% among school-age children [1]. It co-occurs with attention-deficit/hyperactivity disorder (ADHD), autism spectrum disorder (ASD; often referred to as autism), specific learning disorder (SLD), and other neurodevelopmental conditions [2]. Meta-analytic and large-scale clinical studies indicate that approximately 50% of individuals with DCD also meet diagnostic criteria for ADHD [3], approximately 20–30% present with ASD or autistic traits [2,4], and approximately 30–50% have a co-occurring SLD [2,5].

Consequently, it can lead to poorer academic achievement [3], increased anxiety and depression [4], and reduced self-esteem [4,5]. Longitudinal studies tracing children with DCD traits into adolescence have shown that motor difficulties persist in 50–70% of cases [6,7,8]. Blank et al. [2] noted that assessment and intervention studies for adults with DCD are limited compared with those for children. One major constraint is the inconsistency across studies and the restricted methods for evaluating motor difficulties. Education and healthcare professionals also tend to lack familiarity with the adjustments and support required for individuals with DCD [9], and there is a shortage of assessment tools and diagnostic processes for adults with this condition [10].

International studies assessing DCD in young adults and adults predominantly employ the Bruininks–Oseretsky Test of Motor Proficiency, Third Edition [11] and the Movement Assessment Battery for Children, Second Edition [12]. To screen young adults, the Adult Developmental Coordination Disorders/Dyspraxia Checklist (ADC) [10], and the Adolescents and Adults Coordination Questionnaire (AAC-Q) are mainly employed. In the United States (U.S.), assessments using ADC [8] have reported that reduced motor coordination correlates with a lower health-related quality of life [13]. Similarly, surveys using the AAC-Q have reported that approximately 7.0% of respondents are likely to have DCD [14], and, in particular, that avoidance behaviors related to DCD—such as procrastination in daily activities, reduced engagement in academic tasks, and withdrawal from social situations—are associated with lower quality-of-life scores [15].

In both the U.S. and the United Kingdom, legal frameworks (e.g., the ADA Amendments Act and the Equality Act/Disabled Students’ Allowance) formally provide for reasonable accommodations in higher education [16,17,18,19,20]. However, awareness and implementation specific to DCD lag behind more familiar conditions (e.g., dyslexia), and DCD-based assessment-driven accommodations remain limited [21,22]. In Japan, statistics from the Japan Student Services Organization [23] show high proportions of students classified as having psychiatric disorders, chronic illness, ADHD, or ASD, whereas DCD is subsumed under “other neurodevelopmental disorders”. Very few surveys have reported the number of students with DCD or the accommodations they receive, and there is no standardized screening tool for DCD. Due to differences in prevalence, diagnoses, and cultural contexts, direct comparisons with non-Japanese data are difficult. Nonetheless, dyslexia, one of the most common diagnoses of SLD in Western higher education, often overlaps with DCD [22,24]. DCD frequently co-occurs with other neurodevelopmental conditions—particularly ADHD, ASD, and SLD—as shown in epidemiological studies [2,22,25] and acknowledged in DSM-5/DSM-5-TR, which explicitly allows concurrent diagnoses when criteria are met [1,26]. Despite the inclusion of these diagnostic relationships in official criteria, Japanese government agencies and support personnel lack sufficient knowledge and assessment resources to identify and support adults with DCD [9]. In a pilot study involving university students, Yasunaga et al. [27] conducted a web-based survey using the AAC-Q and reported that 9.0% of the participants exhibited DCD traits and perceived difficulties characteristic of ASD and ADHD.

In adults and young adults, DCD is assessed either with performance-based motor tests (e.g., BOT2, MABC2) for clinical/diagnostic use or with self-report screeners (e.g., ADC, AAC-Q) suitable for large-scale screening. Since we screened a large university cohort and examined links with ADHD/ASD-related difficulties, learning support needs, and UPI scores, we used the AAC-Q as the primary instrument. However, the validity of the AAC-Q and the relationship of DCD traits (via the AAC-Q) with mental health [28] and lifestyle factors [29] have not yet been examined.

Against this background, this study aimed to (i) estimate the proportion of university students who screen positive for DCD traits using the AAC-Q, (ii) examine how DCD traits are associated with ADHD and ASD-related difficulties, learning support needs, and mental health difficulties (UPI), and (iii) test whether ADHD/ASD difficulties account for the association between DCD traits and mental health difficulties. Understanding the links between DCD traits, neurodevelopmental, and mental health difficulties in higher education settings may highlight the potential value of early, targeted screening in supporting students’ self-understanding and access to appropriate services.

## 2. Materials and Methods

### 2.1. Participants

This cross-sectional study was conducted in April 2024. All undergraduate and graduate students (*N* = 16,295) of the National University A in Western Japan who had undergone annual health examinations were invited to participate in the study. The online survey was open for one month. Immediately after completing the health examination questionnaire, the students were redirected to a webpage inviting them to participate in the study. A total of 57 students were excluded from the analysis because they provided incomplete questionnaire responses or violated the inclusion criteria (age > 30 years, self-reported visual impairment or chronic illness, unspecified diagnosis, non-regular student status, or data inconsistencies). Consequently, 527 participants were included in the final analysis (Figure 1). Among these 527 participants, 6 students (1.1%) self-reported a prior clinical diagnosis of ADHD and 12 students (2.2%) reported a diagnosis of an autism-spectrum condition; diagnostic status was not used in inferential analyses because the study focused on self-reported trait-level difficulties.

### 2.2. Survey Procedure

After completing the university health survey, the students were presented with an information sheet explaining the purpose of the study, data confidentiality, voluntary nature of participation, the right to withdraw at any time, and consent to publish results anonymously. The submission of the questionnaire was regarded as consent. Participants who agreed with the conditions mentioned above clicked “Proceed to survey” and completed the following instruments—the AAC-Q, the UPI, the ADHD Difficulty Scale, the ASD Difficulty Scale, and the 7-item Learning Difficulty Scale for Postsecondary students (LDSP7) and the 10-item Scale for Childhood Learning Difficulties (SCLD10) (both of which are short versions of the Reading and Writing Support Needs Scale) [30].

### 2.3. Instruments

In this study, “ADHD/ASD/SLD traits” denotes participants whose self-reported difficulty scores exceed the established cut-off, reflecting elevated self-perceived difficulties rather than a clinical diagnosis. Coordination was screened using the AAC-Q [31]; ADHD- and ASD-related difficulties were assessed using the shortform Developmental Disorder Difficulty Scales [32,33].

#### 2.3.1. Original Health-Examination Questionnaire

The university health survey included items on lifestyle, current medical care, physical and mental health status, recent subjective symptoms, physical activity, diet, stress, and sleep habits.

#### 2.3.2. AAC-Q

The AAC-Q [31] is a reliable and valid screening tool for individuals aged 16–30 years, comprising 12 self-report items scored on a 5-point Likert scale (1 = never, 5 = always), with higher scores indicating more severe motor difficulties. One example item (taken from the Japanese version provided by the original developer) is: “Fine-motor activities requiring bimanual coordination are difficult for me”. Following Tal-Saban et al. [31,34], we adopted a total score ≥ 32 as the cut-off for DCD traits, based on percentile-based thresholds established in the original AAC-Q standardization work (*N* = 2379), rather than an SD rule. In our sample (*n* = 527; *M* = 20.72, *SD* = 6.72), a score of 32 for the AAC-Q total corresponded to +1.68 SD, which we report only to indicate its relative position within this dataset and not as a basis for defining the cut-off. In this study, “traits” denote participants whose self-reported difficulty scores exceed an established screening cut-off, indicating elevated self-perceived difficulties rather than a clinical diagnosis. AAC-Q cut-offs (≥32/≥27) originate from the original standardization work; any *SD* values reported in the Results indicate relative position within this sample and were not used to define cut-offs. Previous research had shown excellent internal consistency (Cronbach’s α = 0.88) and good test–retest and discriminant validity [31]; the current sample also showed good internal consistency (Cronbach’s α = 0.81). A Japanese version, supplied by the original developer in 2023 and internally reviewed for linguistic clarity and cultural relevance, was used in the present study. Although the Japanese version used here has not been formally validated in a peer-reviewed publication, a preliminary internal assessment of criterion-related validity was conducted in this sample. The AAC-Q total score showed statistically significant and theoretically consistent correlations with ADHD-related difficulties (*r* = 0.65), ASD-related difficulties (*r* = 0.55), learning difficulties (*r* = 0.48), and mental health symptoms (UPI total: *r* = 0.41) (see Section 3.2). These results are consistent with prior findings using comparable instruments as examples, such as the DCDQ–ARS and M-ABC2–JFE. Here, “DCD traits” denotes a screening-positive result on the AAC-Q and does not represent a formal diagnosis. A formal validation study of the Japanese AAC-Q is currently underway and will be reported separately; the present findings should therefore be considered preliminary.

#### 2.3.3. Developmental Disorder Difficulty Scales

The ADHD Difficulty Scale and ASD Difficulty Scale were employed. These are shortened versions of scales developed to quantify the degree of academic and daily life difficulties that university students experience owing to neurodevelopmental characteristics [32,33]. Although no official cut-off scores were established, scores within the upper 5.0% of the distribution as well as markedly elevated scores on particular subdomains were intended to guide the use of the corresponding subscale items when focusing on specific areas of difficulty.

##### ADHD Difficulty Scale

This shortened scale developed by Takahashi et al. [32,33] extracts items related to ADHD tendencies. Each item is rated on a 4-point scale (0 = not difficult at all, 3 = very difficult). The seven subdomains are sustained attention (one item), inattention (two items), impulsivity (two items), lack of planning (one item), irregular sleep rhythm (one item), clumsiness (two items), and organization (one item). Following Takahashi et al. [32,33], students with total scores ≥1.92 were classified as having ADHD-related difficulties. It has good internal consistency (Cronbach’s α = 0.88) [33]; α was 0.82 in the current sample.

##### ASD Difficulty Scale

Developed by Takahashi et al. [32,33], this is a 13-item version of the original 23-item scale created to shorten administration time and enhance usability in counseling settings. Each item is rated on the same 4-point scale as the ADHD version (0 = not difficult at all, 3 = very difficult). This scale comprises two subdomains—interpersonal difficulties (five items) and autism-related difficulties (eight items). In line with Takahashi et al. [32,33], a total score ≥1.57 was interpreted as indicating ASD-related difficulties. It has good internal consistency (overall Cronbach’s α = 0.88; autism-related subscale α = 0.80; interpersonal subscale α = 0.84) [33]. In the current sample, α was 0.89 overall (autism-related α = 0.82; interpersonal α = 0.84).

#### 2.3.4. Reading and Writing Support Needs Scale

The Reading and Writing Support Needs Scale [30] is used to assess reading- and writing-related difficulties and their underlying factors among university students. The items capture challenges common to students with SLD and dyslexia; higher scores indicate a greater likelihood of SLD-related problems. The full scale comprises 93 items—44 addressing current university difficulties and 49 reflecting elementary school experiences. Two short versions of this scale were used in this study—the LDSP7 and the SCLD10. Items are rated on a 4-point scale (1 = does not apply, 4 = applies very much). In accordance with Takahashi et al. [30], a total score ≥2.5 on either form indicated a need for reading- and writing-related support. For SCLD10, we additionally inspected a stricter exploratory threshold (≥3.05) to align with prior practice; conclusions were unchanged. Internal consistency (Cronbach’s α) for the LDSP7 and SCLD10 were 0.68 and 0.83, respectively [30]. In this study, α was 0.67 for the LDSP7 and 0.76 for the SCLD10.

#### 2.3.5. UPI

The UPI [35] is a questionnaire designed to assess the mental health difficulties of university students, including their worries and anxieties. It asks respondents whether they have experienced each of 60 listed symptoms during the past year (for example, “I have no interest in anything”). Prior to commencing, the following instructions (translated from Japanese) were presented:

“This survey is intended to help you understand and improve your health. Please read the following items numerically. If, at any time during the past year, you have occasionally felt or experienced the stated condition, check ‘Yes’; if not, check ‘No’.”

Responses are dichotomous (“Yes” = 1, “No” = 0). Of the 60 items, four constitute a lie scale and are excluded, leaving 56 items for analysis. A total score of 17 or higher indicates poor mental health [36]. The UPI has demonstrated excellent internal consistency (Cronbach’s α = 0.97) [37]. In this sample, α was 0.92, indicating excellent internal consistency.

### 2.4. Data Analysis

All analyses were conducted using IBM SPSS Statistics 26 software (IBM Corp., Armonk, NY, USA). Students with and without DCD traits were compared using the AAC-Q, ADHD Difficulty Scale, ASD Difficulty Scale, LDSP7, SCLD10, UPI, and mental health items. Chi-square tests were applied to categorical variables, and Shapiro–Wilk tests indicated that the continuous variables were not normally distributed; therefore, non-parametric procedures were applied—Mann–Whitney U tests for group comparisons and Spearman’s rank-order correlations for associations. Scale reliabilities were assessed with Cronbach’s α. Statistical significance was set at *p* < 0.05, and effect sizes for correlations were interpreted according to Cohen’s (1988) [38] conventions (small = |*r*| 0.10, medium = 0.30, large = 0.50); statistical power ≥0.80 was deemed adequate for sample-size planning. To address multiple comparisons for the planned correlation family (number of correlations m = 16), Bonferroni-adjusted *p*-values were computed as *p*_adj = *p* × 16 (values > 1 set to 1), and significance was evaluated at α = 0.05.

Simple linear regressions were used to test whether the AAC-Q scores predicted ADHD difficulties, ASD difficulties, UPI scores, and learning difficulties (LDSP7 scores). Hierarchical multiple regression analysis was used to analyze the predictors of UPI scores. The AAC-Q was entered first, followed by ASD and ADHD difficulties. For each model, standardized β coefficients, *p*-values, and coefficients of determination (R^2^ and adjusted R^2^) are reported. Exploratory models including age and gender as covariates were tested, but neither covariate was significant, and their inclusion did not materially alter parameter estimates or model fit (ΔAIC < 2); therefore, unadjusted models are presented herein. Exploratory mediation analysis was conducted using a nonparametric bootstrap approach (Hayes’ PROCESS macro, Model 4). The AAC-Q total score was specified as the predictor (X), ADHD and ASD difficulty scores as parallel mediators (M1, M2), and the UPI total as the outcome (Y). Age and sex were included as covariates on all paths. Indirect effects were estimated from 5000 bootstrap resamples with bias-corrected 95% confidence intervals; an effect was considered statistically significant when the 95% CI did not include zero. Given the cross-sectional design, these analyses are interpreted as indirect statistical associations rather than causal mechanisms.

### 2.5. Ethical Considerations

This study was approved by the Ethics Committee of the Campus Life Health Support and Counseling Center, The University of Osaka (approval no. 10). The following statement was read to the participants before they started with the web survey:

“Participation is voluntary, and there is no disadvantage if you choose not to respond. Your answers are not reported to the university. All data and personal information will be password-protected, handled only by the principal investigator, and statistically analyzed. No data will be disclosed to third parties, and the survey system will be completely deleted after completion of the study. The results will be published only in an aggregate form, which does not identify individuals. Personal information will be anonymized, and any e-mail correspondence will be promptly deleted after receipt”. Informed consent was obtained from all participants after submission of the completed questionnaire.

## 3. Results

### 3.1. Overall Results

A total of 527 students (mean age = 21.1 ± 2.79 years; 227 women) participated in the study, yielding a valid response rate of 3.2%. Thirty-nine students (7.4%) scored ≥ 32 and 93 students (17.6%) scored ≥ 27 on the AAC-Q. In this sample, 7.4% scored at or above the AAC-Q screening cut-off (≥32). This proportion indexes a screening threshold within our cohort and is not a diagnostic or population prevalence estimate. In the present sample, a score of 32 for the AAC-Q total corresponded to +1.68 SD. On the ADHD Difficulty Scale, 17 students (3.2%) scored ≥ 1.92, whereas 50 students (9.5%) scored ≥ 1.34. On the ASD Difficulty Scale, 37 students (7.0%) scored ≥ 1.57, whereas 106 students (20.1%) scored ≥ 1.06. None of the participants scored ≥ 3.0 on the LDSP7, and 3 students (1.5%) scored ≥ 2.5. For SCLD10, 0 students scored ≥ 2.5; likewise, 0 scored ≥ 3.05. Sixty-three students (12.0%) obtained UPI totals of 17 or higher, indicating greater mental health difficulties.

Table 1 shows group comparisons between students with pronounced DCD traits (AAC-Q ≥ 32) and those without such traits. The DCD-positive group had higher total and subscale scores on the ADHD and ASD Difficulty Scales, LDSP7, SCLD10, and UPI. Significant differences were also found in smoking status (*p* = 0.002, d = 0.134) and weekly evening meal frequency (*p* = 0.022, d = 0.10), whereas weekday sleep duration did not differ significantly (*p* = 0.309, d = 0.04).

### 3.2. Correlations Among AAC-Q, Difficulty Scales, UPI, and Health Survey Items

As shown in Table 2, the strongest association was with ADHD difficulties (*r* = 0.65), followed by ASD difficulties (*r* = 0.55) and LDSP7 (*r* = 0.48); the correlation with UPI total was *r* = 0.41 (all *p* < 0.001). Moderate correlations were found with inattention (*r* = 0.47), lack of planning (*r* = 0.44), impulsivity (*r* = 0.42), interpersonal difficulties (*r* = 0.45), the LDSP7 total (*r* = 0.48), and the UPI total (*r* = 0.41). Bonferroni-adjusted *p*-values were calculated as *p*_adj = *p* × 16 (values >1 set to 1), and statistical significance was evaluated at α = 0.05.

As shown in Table 2, after *p*-value adjustment, most correlations between AAC-Q and the other measures remained significant, except for difficulty in maintaining concentration, which did not reach significance after *p*-value adjustment. Notably, the associations with SCLD10 and perceived stress remained significant after *p*-value adjustment. Weak positive correlations with the UPI suicidal ideation item (*r* = 0.19), the SCLD10 total (*r* = 0.36), and perceived stress (*r* = 0.31) also remained significant after *p*-value adjustment. In contrast, no significant correlations were found with smoking status (*r* = 0.063, *p* = 0.149), weekly evening meal frequency (*r* = 0.06, *p* = 0.161), or weekday sleep duration (*r* = 0.06, *p* = 0.170).

## 3.3. Regression Analyses of the AAC-Q, Difficulty Scales, and UPI Totals

### 3.3.1. Simple Regression Analysis

We used simple linear regression to examine whether the AAC-Q total score predicts ADHD difficulties, ASD difficulties, total UPI, total LDSP7, and perceived stress. As summarized in Table 3, AAC-Q was a significant predictor of all outcomes (all *p* < 0.001; model R^2^ = 0.10–0.43).

### 3.3.2. Multiple Regression Analysis

Hierarchical multiple regression was performed to examine the factors predicting the total UPI score (Table 4). Model 1, which included only the AAC-Q total score, showed that the AAC-Q was a significant predictor (β = 0.43, *p* < 0.001). When the ASD Difficulty Scale total score was added in Model 2, ASD difficulties emerged as a significant predictor (β = 0.62, *p* < 0.001), whereas the AAC-Q remained significant with a smaller standardized coefficient (β = 0.08, *p* = 0.039). In Model 3, after adding the ADHD Difficulty Scale total, ASD difficulties (β = 0.58, *p* < 0.001) and ADHD difficulties (β = 0.13, *p* = 0.006) remained significant predictors of the UPI total, whereas the AAC-Q was no longer significant (β = 0.02, *p* = 0.064). This model explained 45.2% of the variance in UPI scores (adjusted R^2^ = 0.452). For detailed coefficients, standard errors (SEs), confidence intervals (CIs), and model diagnostics, see Appendix A.

### 3.3.3. Mediation Analysis

In an exploratory parallel mediation model (PROCESS Model 4; 5000 bootstrap resamples), the total indirect effect of AAC-Q on UPI via ADHD and ASD difficulties was statistically significant (ab_total = 0.4782, 95% CI [0.3676, 0.5994]). Both specific indirect effects were also significant—via ADHD (ab_1_ = 0.0995, 95% CI [0.0126, 0.1878]) and via ASD (ab_2_ = 0.3787, 95% CI [0.2894, 0.4789]). Consistent with these patterns, the direct effect of AAC-Q on UPI was not significant after accounting for the mediators. In line with the study’s cross-sectional design, these findings are interpreted as indirect statistical associations rather than causal mechanisms.

## 4. Discussion

### 4.1. Overall Overview

To the best of our knowledge, this is the first cross-sectional study conducted in Japan involving university students in higher education institutions with DCD traits, with the aim of clarifying how these traits relate to ADHD, ASD, SLD, and UPI scores. Among the 527 respondents, 39 (7.4%) met the criteria for DCD traits, 17 (3.2%) met the criteria for ADHD traits, 37 (7.0%) met the criteria for ASD traits, and 63 (12.0%) had total UPI scores indicating mental health difficulties. This prevalence (7.4%) (screening-positive) closely matches previous AAC-Q studies of young adults, which have typically reported ~5–8% positive screens—for example, 7% in a U.S. university sample [14] and 9% in a Japanese pilot study [27]. These indicators were significantly associated with each other; students with DCD traits scored significantly higher than those without such traits on the ADHD and ASD Difficulty Scales, learning difficulty scales, and measures of mental health problems. The AAC-Q score showed strong correlations with ADHD and ASD difficulties, and moderate correlations with LDSP7 and UPI scores, suggesting that DCD traits may be linked to multifaceted difficulties. For clarity, we interpret the 7.4% not as a diagnostic boundary or population prevalence, rather as a screen-positive proportion at the AAC-Q cut-off (≥32) based on the original standardization.

These findings indicate that a non-negligible proportion of students in a Japanese university present with DCD traits that are associated with ADHD and ASD characteristics, and that those with neurodevelopmental traits may also experience mental health problems. These findings underscore the importance of considering broad interviews and screenings at university admission as a means to better understand, and potentially support, students with diverse traits. The following sections discuss the relationships among DCD traits, ADHD/ASD difficulties, learning difficulties, and mental health problems, as well as the importance of comprehensive screening for multifaceted difficulties associated with DCD.

### 4.2. Relationships Among DCD Traits, ADHD/ASD Difficulties, Learning Difficulties, and UPI Scores

Group comparisons showed that students with DCD traits reported greater difficulties related to ADHD, ASD, and learning difficulties, as well as more mental health problems. Previous studies have noted that ADHD- and ASD-related traits are often accompanied by mental health issues; the results of this study are consistent with this tendency. These findings indicate that students with DCD traits may face multifaceted challenges and, therefore, require mental health screening and appropriate support. Most moderate-to-strong correlations survived Bonferroni *p*-value adjustment (*p*_adj = *p* × 16; α = 0.05), supporting the robustness of the main association pattern. Furthermore, analyzing AAC-Q as a continuous variable yielded the same pattern of associations, confirming that dichotomization did not bias our conclusions. Our findings should not be interpreted as supporting universal screening. Rather, DCD screening may be most meaningful when conducted alongside ADHD/ASD screening in settings where neurodevelopmental traits frequently cooccur and relate to mental health difficulties.

Simple regression analysis showed that the AAC-Q total score predicted ADHD difficulties (R^2^ = 0.43) with strong accuracy and ASD difficulties (R^2^ = 0.29) with moderate accuracy, and that it was also related to mental health difficulties measured by the UPI [2,39]. This pattern corroborates previous reports of frequent comorbidities among DCD, ADHD, and ASD [2,40,41]. Hierarchical multiple regression analysis showed that the AAC-Q alone predicted total UPI scores, but its effect became non-significant once ADHD and ASD difficulties were added, with these factors becoming the principal predictors of mental health problems (final model R^2^ = 0.46, adjusted R^2^ = 0.452) [42,43]. A bootstrapped mediation analysis confirmed that the direct effect of AAC-Q on UPI was non-significant (B = 0.025, *p* = 0.638), while the total indirect effects via ADHD and ASD difficulties remained significant (total indirect = 0.4782, 95% CI [0.368, 0.599]), supporting the hierarchical regression pattern and indicating that DCD traits influence mental health difficulties primarily via ADHD/ASD-related difficulties. This conclusion aligns with reports that adolescents and adults with DCD are prone to psychological difficulties, such as anxiety and depression [28,40,44,45].

University students aged approximately 18–25 years fall within the “emerging-adulthood” period, during which executive functions centered on the prefrontal cortex are still maturing, and therefore, their vulnerability to stress is elevated [46,47]. In case of students with DCD traits undergoing this developmental transition, assessment and support should address both neurodevelopmental characteristics and mental health. Gabbard et al. [14] emphasized that a thorough evaluation of mental health is indispensable when designing support methods for individuals with DCD.

### 4.3. Multiple Difficulties Associated with DCD Traits and the Importance of Comprehensive Screening

The findings demonstrate that DCD traits are linked not only to motor problems but also to cognitive and emotional difficulties, indicating that a DCD perspective is crucial for understanding the multifaceted challenges students encounter in university life. Nevertheless, DCD is often subsumed under SLD [48,49]. Furthermore, compared with ASD and ADHD, recognition, diagnostic frameworks, and support systems in educational and clinical settings remain inadequate. Wilson et al. [21] and Yasunaga et al. [27] have noted that the absence of DCD-specific diagnostic criteria and trained assessors may have contributed to delayed or missed support. Moreover, because individuals with DCD are often unaware of their condition and their complaints extend beyond motor issues [8,49], they may fail to access assistance. Tal-Saban et al. [34] reported that only 8% of adults with DCD cited gross-motor difficulties as their primary concern, whereas 55% identified executive-function difficulties and 53% reported psychosocial difficulties.

In the present study, AAC-Q scores correlated significantly with the ADHD Difficulty Scale subdomains of inattention, impulsivity, and lack of planning, demonstrating non-motor difficulties. These results accord with recent adult studies that document executive-function challenges outside the motor domain [41,42]. Reasonable accommodations for DCD traits—such as extended testing time, provision of notetakers, and computer use—have been reported [22,50]. In contrast, accommodation systems in Japanese higher education institutions are primarily applied to students with ADHD, ASD, and psychiatric disorders [23]; DCD-specific screening and support remain limited in both policy and practice. Collecting DCD screening information at university entry, alongside other neurodevelopmental and mental health assessments, may clarify students’ multifaceted needs at an early stage. These data can then inform the design of academic and mental health support that could mitigate secondary problems and foster students’ self-understanding [51]. We do not propose universal DCD screening; rather, DCD screening is implemented in a targeted manner based on prespecified neurodevelopmental triggers. Specifically, we outline a concise, outcome-based twostep process: (i) a universal online intake at matriculation, or when students proactively report neurodevelopmental difficulties or request accommodations, limited to abbreviated ADHD/ASD difficulty scales and a brief mental health indicator to identify candidates for further assessment; and (ii) targeted DCD screening (AAC-Q) only for students who meet these triggers (e.g., a positive ADHD/ASD screen, self-reported coordination difficulties, academic/clinical referral, or pertinent history), followed—if the student wishes—by an optional multidisciplinary interview to discuss reasonable accommodations and semester-based follow-up. Aggregated results are fed back to disability support offices to refine and expand services, thereby enhancing institutional quality assurance and ensuring equitable learning opportunities for all students.

### 4.4. Limitations

This study had several limitations. First, the participants were restricted to students enrolled at National University A, a single national institution, and most of them were first-year undergraduates. As the overall sample, especially the subgroup with DCD traits, contained a higher proportion of men, the generalizability of the findings is limited. Furthermore, the overall survey response rate was 3.2%, which raises the possibility of nonresponse and self-selection bias; respondents may differ systematically from nonrespondents in neurodevelopmental traits, mental health, or study engagement. Since auxiliary information on nonrespondents was unavailable, we could not implement weighting or other nonresponse adjustments to quantify or correct such bias. Second, the assessment of DCD traits, ADHD, ASD difficulties, and reading–writing support needs relied exclusively on self-reported questionnaires. In particular, the AAC-Q includes non-motor and participation items and does not directly measure motor coordination, as its cut-off scores are based on overseas norms. The AAC-Q is a screening instrument; cutoffs are derived from non-Japanese norms, and potential misclassification cannot be ruled out. Local validation of thresholds is warranted before any prevalence-level inference. Third, its cross-sectional design precludes causal inferences. The elucidation of causal mechanisms requires longitudinal research and analytical approaches, such as structural equation modeling. Finally, although this study examined the associations between DCD traits and lifestyle factors (smoking, diet, and sleep duration), only some differences were statistically significant, and no clear correlations emerged. Future studies should investigate these relationships by using more diverse samples and detailed lifestyle measures. Furthermore, sensitivity analyses indicated that adjusting for age and gender did not change the direction or significance of the results, supporting the robustness of our findings despite these limitations. Future studies could explore whether individuals with elevated DCD traits but minimal ADHD/ASD characteristics exhibit distinct psychosocial profiles.

## 5. Conclusions

A significant proportion of students in Japanese higher education institutions demonstrate traits of DCD, which is associated with neurodevelopmental characteristics and mental health difficulties. Screening using the AAC-Q predicted ADHD-related difficulties with strong accuracy and ASD-related difficulties with moderate accuracy; both sets of difficulties were significant predictors of mental health status as measured by the UPI. Integrating DCD screening into broader assessments of neurodevelopmental and mental health difficulties may facilitate the identification of diverse traits, and inform the development of more appropriate support systems, which could help to mitigate subsequent mental health difficulties. In practice, these results argue for targeted DCD screening implemented alongside ADHD/ASD screening at university entry, rather than universal screening of all students.

### Future Research

Future studies should examine the construct and criterion validity of the Japanese AAC-Q scores, establish cut-off values, and confirm its practical utility in Japan. Research should also involve multiple Japanese higher education institutions and include students of diverse genders, academic years, and majors to explore the relationships between DCD traits, neurodevelopmental difficulties, and UPI scores and to compare students with and without neurodevelopmental traits. In addition, semi-structured interviews with students who exhibit DCD traits should be conducted to investigate social barriers related to DCD and the learning environment, thereby deepening the understanding of the diversity and individuality of support needs.

## Figures and Tables

**Figure 1 brainsci-15-00895-f001:**
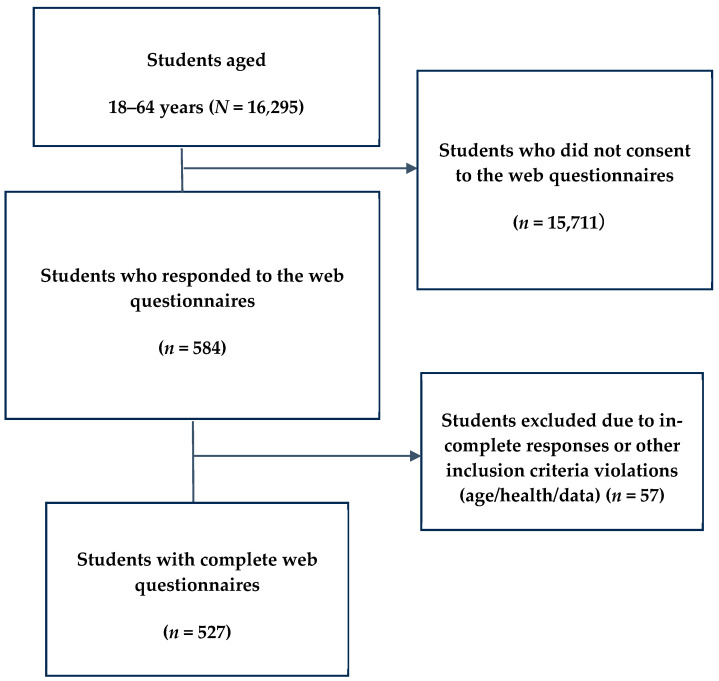
Flow chart of survey data collection. Exclusion criteria: ≥30 years of age, visual impairment, physical disability, chronic illness, or other medical conditions.

**Table 1 brainsci-15-00895-t001:** Comparison of Assessment Results between Groups with and without DCD Traits.

Outcome Measure	Group with DCD Traits		Group Without DCD Traits			
	(*n* = 39)		(*n *= 488)			
	Mean	Median	Mean	Median	*p*-value	Effect size
	(*SD*)	(25th–75th percentile)	(*SD*)	(25th–75th percentile)		
Sex	1.59	2.0 (1.0–2.0)	1.42	1 (1.0–2.0)	0.001 ^†^	0.14
Age	20.92	20 (19–22)	21.07	21 (18–23)	0.808 ^‡^	0.11
AAC-Q						
Total score	36.18	35 (33.0–37.0)	19.48	19 (15.0–23.0)	0.000 ^‡^	0.45
ADHD						
Total score	14.41	13 (10.0–20.0)	5.32	4 (2.0–7.0)	0.000 ^‡^	0.36
Concentration	0.69	1 (0–1.0)	0.77	1 (0–1.0)	0.000 ^‡^	0.03
Inattention	3.05	3 (1.0–5.0)	0.91	0 (0–2.0)	0.000 ^‡^	0.29
Impulsivity	2.77	3 (1.0–4.0)	0.69	0 (0–1.0)	0.000 ^‡^	0.33
Sleep	1.28	1 (0–2.0)	0.64	0 (0–1.0)	0.000 ^‡^	0.18
Planning	1.54	2 (0–3.0)	0.57	0 (0–1.0)	0.000 ^‡^	0.24
Clumsiness	3.33	3 (2.0–5.0)	1.01	1 (0–2.0)	0.000 ^‡^	0.33
Organization	1.74	2 (1.0–3.0)	0.72	1 (0–1.0)	0.000 ^‡^	0.25
ASD						
Total score	16.92	17 (9.0–23.0)	7.43	6 (2.0–11.0)	0.000 ^‡^	0.27
Autism	10.49	10 (4.0–16.0)	4.24	3 (1.0–6.0)	0.000 ^‡^	0.27
Interpersonal	6.44	7 (3.0–9.0)	3.19	2 (1.0–5.0)	0.000 ^‡^	0.2
SLD						
LDSP7 total score	12.03	11 (9.0–14.0)	9.30	9 (7.0–11.0)	0.000 ^‡^	0.22
SCLD10 total score	15.10	14 (11.0–18.0)	12.78	11 (10.0–14.0)	0.001 ^‡^	0.14
UPI						
Total Score	15.00	14 (5.0–21.0)	6.97	5 (1.0–10.0)	0.000 ^‡^	0.23
suicidal ideation	0.28	0.00	0.06	0.00	0.000 ^‡^	0.23
Mental Health						
stress	2.67	3 (2.0–3.0)	2.10	2 (2.0–2.0)	0.000 ^‡^	0.18

AAC-Q: Adolescents and Adults Coordination Questionnaire; ADHD: Attention Deficit Hyperactivity Disorder Difficulty Scale (Short Version); ASD: Autism Spectrum Disorder Difficulty Scale (Short Version); LDSP7: Learning Difficulty Scale for Postsecondary Students (7-item); SCLD10: Scale for Childhood Learning Difficulties (10-item); UPI: University Personality Inventory; ^†^ chi-square test; ^‡^ Mann–Whitney U test. Effect size = r.

**Table 2 brainsci-15-00895-t002:** Relationship between AAC-Q and ADHD, ASD, SLD, UPI, and Mental Health (n = 527).

	1	2	3	4	5	6	7	8	9	10	11	12	13	14	15	16
1. AAC-Q-TS	―															
2. ADHD-TS	**0.65 ****	―														
	[0.59, 0.69]															
3. ADHD-CO	0.04	0.22 **	―													
	[−0.05, 0.12]	[0.13, 0.30]														
4. ADHD-INA	*0.47 ***	0.71 **	−0.02	―												
	[0.40, 0.53]	[0.67, 0.75]	[−0.11, 0.07]													
5. ADHD-IM	*0.42 ***	0.71 **	0.01	0.39 **	―											
	[0.34, 0.49]	[0.67, 0.75]	[−0.08, 0.10]	[0.31, 0.46]												
6. ADHD-PL	*0.44 ***	0.64 **	−0.01	0.44 **	0.37 **	―										
	[0.36, 0.51]	[0.58, 0.69]	[−0.10, 0.08]	[0.37, 0.51]	[0.29, 0.44]											
7. ADHD-SL	0.36 **	0.53 **	0.01	0.22 **	0.28 **	0.35 **	―									
	[0.28, 0.43]	[0.46, 0.59]	[−0.07, 0.10]	[0.14, 0.30]	[0.20, 0.36]	[0.27, 0.43]										
8. ADHD-CL	**0.51 ****	0.77 **	0.01	0.51 **	0.72 **	0.42 **	0.26 **									
	[0.44, 0.57]	[0.73, 0.80]	[−0.08, 0.10]	[0.44, 0.57]	[0.67, 0.76]	[0.34, 0.49]	[0.18, 0.34]									
9. ADHD-OT	*0.48 ***	0.62 **	−0.03	0.45 **	0.30 **	0.39 **	0.37 **	0.31 **								
	[0.41, 0.54]	[0.56, 0.67]	[−0.12, 0.06]	[0.38, 0.52]	[0.22, 0.38]	[0.31, 0.46]	[0.29, 0.44]	[0.23, 0.39]								
10. ASD-TS	**0.55 ****	0.56 **	−0.04	0.32 **	0.50 **	0.33 **	0.35 **	0.54 **	0.34 **							
	[0.48, 0.61]	[0.49, 0.61]	[−0.13, 0.04]	[0.24, 0.40]	[0.43, 0.57]	[0.25, 0.41]	[0.27, 0.42]	[0.47, 0.60]	[0.26, 0.42]							
11. ASD-AU	**0.53 ****	0.57 **	−0.04	0.35 **	0.52 **	0.32 **	0.35 **	0.54 **	0.34 **	0.93 **						
	[0.47, 0.59]	[0.51, 0.63]	[−0.12, 0.05]	[0.27, 0.43]	[0.46, 0.58]	[0.24, 0.40]	[0.27, 0.43]	[0.48, 0.60]	[0.26, 0.42]	[0.92, 0.94]						
12. ASD-INT	*0.45 ***	0.42 **	−0.04	0.21 **	0.38 **	0.28 **	0.28 **	0.43 **	0.28 **	0.88 **	0.66 **					
	[0.38, 0.52]	[0.35, 0.49]	[−0.13, 0.04]	[0.13, 0.29]	[0.30, 0.45]	[0.20, 0.36]	[0.20, 0.36]	[0.36, 0.50]	[0.20, 0.36]	[0.86, 0.90]	[0.61, 0.71]					
13. LDSP 7-TS	*0.48 ***	0.44 **	−0.02	0.28 **	0.39 **	0.30 **	0.23 **	0.43 **	0.30 **	0.43 **	0.43 **	0.34 **				
	[0.42, 0.55]	[0.36, 0.51]	[−0.11, 0.07]	[0.20, 0.36]	[0.31, 0.46]	[0.21, 0.38]	[0.15, 0.32]	[0.36, 0.50]	[0.22, 0.38]	[0.36, 0.50]	[0.36, 0.50]	[0.26, 0.41]				
14. SCLD10-TS	0.36 **	0.34 **	−0.03	0.21 **	0.28 **	0.27 **	0.17 **	0.33 **	0.21 **	0.40 **	0.38 **	0.33 **	0.60 **			
	[0.28, 0.43]	[0.26, 0.42]	[−0.12, 0.06]	[0.13, 0.30]	[0.20, 0.36]	[0.19, 0.35]	[0.09, 0.26]	[0.28, 0.40]	[0.12, 0.30]	[0.32, 0.47]	[0.30, 0.45]	[0.25, 0.41]	[0.54, 0.65]			
15 UPI-TS	*0.41 ***	0.45 **	0.03	0.28 **	0.33 **	0.28 **	0.33 **	0.42 **	0.27 **	0.60 **	0.60 **	0.48 **	0.29 **	0.25 **		
	[0.33, 0.48]	[0.38, 0.52]	[−0.06, 0.12]	[0.20, 0.36]	[0.25, 0.41]	[0.20, 0.36]	[0.25, 0.41]	[0.35, 0.49]	[0.19, 0.35]	[0.54, 0.66]	[0.54, 0.66]	[0.41, 0.55]	[0.21, 0.37]	[0.16, 0.33]		
16 UPI-SI	0.19 **	0.24 **	0.03	0.14 **	0.19 **	0.20 **	0.20 **	0.23 **	0.07	0.28 **	0.29 **	0.22 **	0.16 **	0.14 **	0.32 **	
	[0.10, 0.27]	[0.15, 0.32]	[−0.06, 0.12]	[0.06, 0.23]	[0.10, 0.27]	[0.12, 0.29]	[0.11, 0.28]	[0.15, 0.32]	[−0.02, 0.16]	[0.19, 0.35]	[0.21, 0.37]	[0.13, 0.30]	[0.08, 0.25]	[0.06, 0.24]	[0.23, 0.39]	
17 MT-FS	0.30 **	0.30 **	−0.00	0.21 **	0.23 **	0.18 **	0.25 **	0.29 **	0.16 **	0.42 **	0.43 **	0.35 **	0.12 **	0.11 *	0.54 **	0.38 **
	[0.22, 0.38]	[0.21, 0.37]	[−0.09, 0.09]	[0.12, 0.29]	[0.15, 0.32]	[0.10, 0.27]	[0.17, 0.32]	[0.21, 0.37]	[0.07, 0.24]	[0.35, 0.50]	[0.36, 0.50]	[0.27, 0.42]	[0.03, 0.20]	[0.02, 0.19]	[0.47, 0.60]	[0.30, 0.45]

* *p* < 0.05. ** *p* < 0.01. [ ] indicate the 95% confidence intervals. Formatting denotes effect size for correlations with AAC-Q only: bold, large (|*r*| ≥ 0.50); italics, medium (0.30 ≤ |*r*| < 0.50). Abbreviations (Table 2 only): MT = mental health; TS = total score; CO = concentration; INA = inattention; IM = impulsivity; PL = planning; SL = sleep; CL = clumsiness; OT = organization and tidiness; AU = autism; INT = interpersonal; SI = suicidal ideation; FS = feeling stressed. Bonferroni-adjusted *p*-values were computed (*p*_adj = *p* × 16; maximum 1), and significance was assessed at α = 0.05 for the planned 16 correlations involving AAC-Q.

**Table 3 brainsci-15-00895-t003:** Simple Regression Analyses Predicting ADHD, ASD, UPI, LDSP7, and Perceived Stress from the AAC-Q Total Score.

Dependent Variable	Predictor	β	R^2^	Adjusted R^2^	*p*-Value
ADHD Total Score	AAC-Q Total Score	0.653	0.427	0.426	*p* < 0.001 **
ASD Total Score	AAC-Q Total Score	0.541	0.293	0.291	*p* < 0.001 **
UPI total	AAC-Q Total Score	0.417	0.174	0.172	*p* < 0.001 **
LDSP7 total	AAC-Q Total Score	0.474	0.225	0.223	*p* < 0.001 **
Perceived stress	AAC-Q Total Score	0.321	0.103	0.101	*p* < 0.001 **

β: standardized coefficient; AAC-Q: Adolescents and Adults Coordination Questionnaire; ADHD: attention deficit hyperactivity disorder; ASD: autism spectrum disorder; ** *p* < 0.001, *p* < 0.05.

**Table 4 brainsci-15-00895-t004:** Hierarchical Multiple Regression Analyses Predicting the UPI Total.

Variables	B	Standard Error (SE)	Standardized β	t-Value	*p*-Value	95% CI	Variance Inflation Factor (VIF)
Constant	0.736	0.903	―	0.815	0.415	[−3.546, 5.018]	-
AAC-Q Total Score	0.025	0.053	0.021	0.471	0.638	[−0.208, 0.258]	1.861
ASD Total Score	0.632	0.045	0.58	13.976	<0.001 **	[0.543, 0.721]	1.652
ADHD Total Score	0.196	0.072	0.126	2.737	0.006 *	[0.055, 0.336]	2.039

Model Statistic. Coefficient of determination: R^2^ = 0.456, Adjusted R^2^ = 0.452, F (3, 523) = 145.887, *p* < 0.001. Durbin-Watson value = 1.907 (no autocorrelation detected). Β: standardized coefficient; CI: confidence interval; AAC-Q: Adolescents and Adults Coordination Questionnaire; ADHD: attention deficit hyperactivity disorder; ASD: autism spectrum disorder; ** *p* < 0.001, * *p* < 0.01, *p* < 0.05.

## Data Availability

The study is ongoing. To protect the originality and accuracy of this research, the raw data that support the conclusions of this article will be made available by the authors upon reasonable request.

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
