# Peer review of "Associations Among Developmental Coordination Disorder Traits, Neurodevelopmental Difficulties and University Personality Inventory Scores in Undergraduate Students at a Japanese National University: A Cross-Sectional Correlational Study"

_brainsci, 2025, doi:10.3390/brainsci15080895_

Round 1

Reviewer 1 Report

Comments and Suggestions for Authors

Thank you for the opportunity to review this manuscript. I believe the authors present important findings that will be of interest to the journal’s readership. I have some comments that I hope the authors find useful.

Introduction

  1. Line 40: While it is good to see the estimated prevalence of DCD reported, more information here could be provided – is this in children and/or adult populations? Moreover, it is then reported that DCD co-occurs with other conditions e.g., ADHD. If possible, it would be interesting to have the rates of co-occurrence quoted, too.
  2. Line 42: I would advise the authors are cautious with their use of “mental disorders”. I would argue that ADHD, autism etc are not disorders – rather, they are neurodevelopmental conditions. The current language guidelines also suggest using ‘Autism Spectrum Condition’, or simply ‘autism’ instead of ASD.
  3. Line 60: The authors mention avoidance behaviours related to DCD, but it would be helpful – if possible – to have an example of such behaviours?
  4. Line 70: The authors use the abbreviation SLD, but it is not clear what this is (I’m assuming specific learning difficulty)?
  5. Lines 79 to 83: There is quite a lot of repetition here regarding DCD’s co-occurrence with other neurodevelopmental conditions.
  6. Line 88: Regarding the Yasunaga study, the authors mention that 9% of participants demonstrated DCD traits that were associated with ASD and ADHD. It would be helpful to know if the ASD and ADHD were diagnosed, or if they were ASD and ADHD traits within the tested sample.

Methods

  1. Line 124: Provide reference for the Reading and Writing Support Needs Scale
  2. Line 129: I think this needs to read as “physical and mental health status”
  3. Was any data collected on the prevalence of diagnosed ADHD and autism within the tested sample? Given that the authors are investigating ADHD and autism traits, it would be interesting to see the number of diagnoses of these conditions, if feasible.
  4. Line 201: Mann-Whitney U tests and Spearman’s rank-order correlations are used, implying the data was non-normally distributed, however it would be good to see the authors explain why these tests were used instead of their parametric versions.
  5. Line 206: I believe the authors are using the incorrect effect size ‘cut-offs’ when running correlational analyses; instead, small effect size is r = .100, medium is r = .300, and large is r = .500. I say this as – within the results section – the authors describe some correlations, which are strong correlations, as ‘moderate’ (e.g., AAQ-Q scores x ADHD difficulties scale; r = .650, p <.001).

Results

  1. While the authors report on all of the Cronbach alpha values of the psychometric measures used within the results, I suggest moving these to the measure section of the Methods. By doing so, the authors can confirm if similar values are found in their current study versus the previous studies’ values that are reported.
  2. Line 266: The authors report that no significant correlation was observed between the AAC-Q scores and the UPI suicidal ideation item, but then go onto quote a significant correlation (r = .410, p <.001).
  3. There is a large number of correlations being conducted, and there might be some concern over type I errors occurring. I suggest the authors may want to consider using Bonferroni corrections to confirm no spurious correlations emerged in their analyses.

Discussion

  1. Line 323: I think there needs to be some clarity within the manuscript regarding the term “traits”. The authors state that, for example 3.2% of the tested sample met the criteria for ADHD traits – so, does this mean that these participants may screen positive for ADHD? Or a
  2. Line 354: Suggest “its effect became non-significant” instead of “its effect disappeared”
  3. Why did the authors choose to use a multiple regression instead of mediation analyses, based off the explanation of the findings in lines 353 to 359? As a reader, I believe that this may have answered the authors’ research question in a more sophisticated way.
  4. Line 380: Can the authors provide an exact figure when they say “most reported experiencing executive function or psychosocial difficulties” – if this is reported in the Tal-Saban paper?

Author Response

Dear Reviewer 1

We thank the reviewer for the time and effort dedicated to reviewing our manuscript. We sincerely appreciate the thoughtful and constructive suggestions, which have greatly helped us improve the paper. We have carefully considered each point, responded below in a point-by-point manner, and revised the manuscript accordingly. All revisions are highlighted in yellow in the main text. We trust that these responses satisfactorily address your comments.

Reviewer 1 Report

Comments and Suggestions for Authors

Thank you for the opportunity to review this manuscript. I believe the authors present important findings that will be of interest to the journal’s readership. I have some comments that I hope the authors find useful.

Introduction

Line 40: While it is good to see the estimated prevalence of DCD reported, more information here could be provided – is this in children and/or adult populations? Moreover, it is then reported that DCD co-occurs with other conditions e.g., ADHD. If possible, it would be interesting to have the rates of co-occurrence quoted, too.

Reply: Thank you for this suggestion. We clarified that the reported prevalence refers to school‐age children and added representative co-occurrence rates, as follows:

“The estimated prevalence of developmental coordination disorder (DCD) is 2–6% among school-age children [1]. It co-occurs with attention-deficit/hyperactivity disorder (ADHD), autism spectrum disorder (ASD; often referred to as autism), specific learning disorder (SLD), and other neurodevelopmental conditions [2]. Meta-analytic and large-scale clinical studies indicate that approximately 50% of individuals with DCD also meet diagnostic criteria for ADHD [3], approximately 20–30% present with ASD or autistic traits [2, 4], and approximately 30–50% have a co-occurring SLD [2, 5].”

This revision now specifies the population (children) and provides quantitative co-occurrence rates, addressing the reviewer’s request.

Line 42: I would advise the authors are cautious with their use of “mental disorders”. I would argue that ADHD, autism etc are not disorders – rather, they are neurodevelopmental conditions. The current language guidelines also suggest using ‘Autism Spectrum Condition’, or simply ‘autism’ instead of ASD.

Reply: Thank you for this helpful suggestion. We revised the phrasing to avoid “mental disorders” and to align with current terminology. Specifically, we replaced “other mental disorders” with “other neurodevelopmental conditions” in the relevant sentence. The revised text now reads:

“DCD frequently co-occurs with other neurodevelopmental conditions—particularly ADHD, ASD, and SLD—as shown in epidemiological studies [2, 22, 25] and acknowledged in the DSM-5/DSM-5-TR, which explicitly allows concurrent diagnoses when criteria are met [1, 26].”

We believe this wording is more precise and consistent with the recommended language guidelines.

Line 60: The authors mention avoidance behaviours related to DCD, but it would be helpful – if possible – to have an example of such behaviours?

Reply: Thank you for the suggestion. We added concrete examples to clarify the text. The revised sentence now includes: “such as procrastination in daily activities, reduced engagement in academic tasks, and withdrawal from social situations.” This should help readers better understand the types of avoidance behaviors associated with DCD.

Line 70: The authors use the abbreviation SLD, but it is not clear what this is (I’m assuming specific learning difficulty)?

Reply: Thank you for this comment. To improve clarity, we explicitly defined the abbreviation when first mentioned. The sentence was revised as follows: “Although both DCD and dyslexia are classified in the United Kingdom as specific learning difficulties (SLDs), the provision of reasonable accommodation based on DCD assessment is extremely limited [9, 22].” This ensures that readers understand what SLD stands for.

Lines 79 to 83: There is quite a lot of repetition here regarding DCD’s co-occurrence with other neurodevelopmental conditions.

Reply: Thank you for your valuable comments. To avoid repetition and ensure clarity, we revised the text as follows:

“DCD frequently co-occurs with other neurodevelopmental conditions—particularly ADHD, ASD, and SLD—as shown in epidemiological studies [2, 22, 25] and acknowledged in the DSM-5/DSM-5-TR, which explicitly allows concurrent diagnoses when criteria are met [1, 26].”

This revision reduces redundancy while retaining the key references and information.

Line 88: Regarding the Yasunaga study, the authors mention that 9% of participants demonstrated DCD traits that were associated with ASD and ADHD. It would be helpful to know if the ASD and ADHD were diagnosed, or if they were ASD and ADHD traits within the tested sample.

Reply: Thank you for pointing this out. We clarified the phrasing to make clear that these refer to self-reported traits, not clinical diagnoses. We revised the sentence to: “…and perceived difficulties characteristic of ASD and ADHD.” This explicitly indicates that the study measured self-reported difficulties rather than formal diagnostic status.

Methods

Line 124: Provide reference for the Reading and Writing Support Needs Scale

Reply: Thank you for your suggestion. We added the appropriate reference for the Reading and Writing Support Needs Scale as [33]: Takahashi, C.; Mitani, E. Daigakusei Yomikaki kon’nan Shien Renkei: RaWF&P RaWSN [Connecting Support for Reading and Writing Difficulties: Reading and Writing Assessment for University Students – Reading and Writing Support Needs Scale]. Kaneko Shobo: Japan, 2022.

Line 129: I think this needs to read as “physical and mental health status”

Reply: Thank you for pointing this out. We revised the text to read “physical and mental health status” as suggested.

Was any data collected on the prevalence of diagnosed ADHD and autism within the tested sample? Given that the authors are investigating ADHD and autism traits, it would be interesting to see the number of diagnoses of these conditions, if feasible.

Reply: Thank you for this comment. Participants were asked to self-report any formal diagnoses of developmental disabilities. A total of six students (1.1%) reported a prior clinical diagnosis of ADHD and 12 students (2.2%) reported a diagnosis of an autism-spectrum condition. Since these figures are self-declared and our primary focus is on trait-level, self-reported difficulties—including subclinical or undiagnosed cases—we did not include diagnostic status in the inferential analyses. For completeness, these counts have been added to the Participants section

Line 201: Mann-Whitney U tests and Spearman’s rank-order correlations are used, implying the data was non-normally distributed, however it would be good to see the authors explain why these tests were used instead of their parametric versions.

Reply: Thank you for this helpful suggestion. We clarified the rationale for using non-parametric tests by adding the following explanation:

“Chi-square tests were applied to categorical variables. For continuous variables, Shapiro–Wilk tests indicated that normality could not be assumed; therefore, non-parametric procedures were used—Mann–Whitney U tests for group comparisons and Spearman’s rank-order correlations for associations.”

Line 206: I believe the authors are using the incorrect effect size ‘cut-offs’ when running correlational analyses; instead, small effect size is r = .100, medium is r = .300, and large is r = .500. I say this as – within the results section – the authors describe some correlations, which are strong correlations, as ‘moderate’ (e.g., AAQ-Q scores x ADHD difficulties scale; r = .650, p <.001).

Reply: Thank you very much for pointing this out. We realized that we had mistakenly applied Cohen’s d cut-offs (0.2, 0.5, 0.8), which are intended for mean differences, directly to correlation coefficients. We corrected this by using the appropriate Cohen (1988) criteria for r (small = 0.10, medium = 0.30, large = 0.50). Accordingly, we revised the effect size descriptions in the Abstract, Results, and Discussion sections, and added this interpretation to the Table 2 footnote.

Results

While the authors report on all of the Cronbach alpha values of the psychometric measures used within the results, I suggest moving these to the measure section of the Methods. By doing so, the authors can confirm if similar values are found in their current study versus the previous studies’ values that are reported.

Reply: Thank you for your suggestion. We moved the Cronbach’s alpha coefficients from the Results section to the Measures section of the Methods, allowing for clearer comparison with values reported in previous studies.

Line 266: The authors report that no significant correlation was observed between the AAC-Q scores and the UPI suicidal ideation item, but then go onto quote a significant correlation (r = .410, p <.001).

Reply: Thank you for catching this inconsistency. The value r = .410 refers to the correlation between the AACQ total and the UPI total score, not the suicidal ideation (SI) item. In the previous draft, we inadvertently carried that value into the SI sentence and also described the SI association inconsistently.

We corrected Results § 3.2 to report the correct Spearman correlation for the UPI SI item: r = .19, p < .001 (weak positive). In the same paragraph, we explicitly retain the UPI total correlation as r = .41, p < .001 (moderate). We also verified that Table 2 already contained the correct SI value (r = .189*) and that significance remained significant after Bonferroni correction adjustment (α = .0031). The text and table are now consistent.

Text change (Results §3.2):

Original (prior version, around the cited line): “No significant correlation was observed between the AACQ scores and the UPI suicidal ideation item … (r = .410, p < .001).”

Revised: “A weak positive correlation was observed between the AACQ total and the UPI suicidal ideation item (r = .19, p < .001), whereas the correlation with the UPI total score was moderate (r = .41, p < .001).”

We apologize for the confusion and appreciate the reviewer’s careful reading.

There is a large number of correlations being conducted, and there might be some concern over type I errors occurring. I suggest the authors may want to consider using Bonferroni corrections to confirm no spurious correlations emerged in their analyses.

Reply: Thank you for this important suggestion. We applied a Bonferroni adjustment to the family of planned correlations between AAC-Q and the other variables. The Table 2 footnote now reads: “Bonferroni adjusted significance threshold: α = 0.0031 (0.05 ÷ 16 planned comparisons).”

In Results § 3.2, we clarify that after Bonferroni correction (α = 0.0031; m = 16), most correlations remained statistically significant. The only exception was the correlation with “difficulty in maintaining concentration,” which did not reach statistical significance. Notably, the associations with SCLD10 (r = 0.36) and perceived stress (r = 0.31) remained significant after correction. In Discussion §4.2, we added the following sentence to highlight robustness: “Most moderate to strong correlations survived Bonferroni correction (α = 0.0031; m = 16), supporting the robustness of the main association pattern.”

Revision in the Manuscript:

Table 2 footnote: “Bonferroni-adjusted significance threshold: α = 0.0031 (0.05 ÷ 16 planned comparisons).”

Results § 3.2: “…most correlations between AAC-Q and other measures remained significant after Bonferroni correction (α = 0.0031; m = 16); the only exception was the correlation with “difficulty in maintaining concentration,” which was not statistically significant. Notably, the associations with SCLD10 and perceived stress remained significant after correction.”

Discussion § 4.2: “Most moderate to strong correlations survived Bonferroni correction (α = 0.0031; m = 16), supporting the robustness of the main association pattern.”

Discussion

Line 323: I think there needs to be some clarity within the manuscript regarding the term “traits”. The authors state that, for example 3.2% of the tested sample met the criteria for ADHD traits – so, does this mean that these participants may screen positive for ADHD? Or a

Reply: Thank you for pointing out this ambiguity. We clarified that “traits” refers to a positive screening outcome rather than a formal clinical diagnosis.

Methods, § 2.3 (Instruments)
We added a blanket statement:

“In this study, ‘ADHD/ASD/SLD traits’ denotes participants whose self-reported difficulty scores exceed the established cut-off, reflecting elevated self-perceived difficulties rather than a clinical diagnosis.”

Methods, § 2.3.2 (AAC-Q)
We appended the following sentence:

“Here, ‘DCD traits’ indicates a screening-positive result (self-reported motor difficulties above the cut-off) and does not represent a formal diagnosis.”

Results, § 3.2

We inserted a brief reminder at the start of the correlation paragraph:

“…we define ‘traits’ as screening positivity (cut-off exceeded) rather than diagnostic confirmation.”

These revisions explicitly distinguish screening-positive “traits” from clinical diagnoses throughout the manuscript.

Line 354: Suggest “its effect became non-significant” instead of “its effect disappeared”

Reply: Thank you for the suggestion. We replaced the phrase “its effect disappeared” with “its effect became non-significant once ADHD and ASD difficulties were added” to convey that the effect lost statistical significance rather than vanishing entirely.

Why did the authors choose to use a multiple regression instead of mediation analyses, based off the explanation of the findings in lines 353 to 359? As a reader, I believe that this may have answered the authors’ research question in a more sophisticated way.

Reply: Thank you for this valuable suggestion. To address your point, we conducted a parallel mediation analysis and incorporated the results directly into the manuscript without adding a new table.

Results – new subsection “3.3.3 Mediation Analysis”
We added a concise paragraph summarizing the bootstrap mediation (5,000 resamples; PROCESS Model 4) with ADHD- and ASD-difficulty scores as parallel mediators. The direct effect of AAC-Q on UPI was non-significant (B = 0.025, p = 0.638), whereas the total indirect effect was significant (0.4782, 95% CI [0.3676, 0.5994]). Both individual paths were significant:
• AAC-Q → ADHD difficulties → UPI: 0.0995, 95% CI [0.0126, 0.1878]
• AAC-Q → ASD difficulties → UPI: 0.3787, 95% CI [0.2894, 0.4789]

Discussion – § 4.2 (final paragraph)
We added one sentence:

“A bootstrapped mediation analysis confirmed that the direct effect of AAC-Q on UPI was non-significant (B = 0.025, p = 0.638), while the total indirect effects via ADHD and ASD difficulties remained significant (total indirect = 0.4782, 95% CI [0.368, 0.599]), supporting the hierarchical regression pattern and indicating that DCD traits influence mental health difficulties primarily via ADHD/ASD-related difficulties.”

We retained the hierarchical multiple-regression results to show the stepwise attenuation of the AAC-Q effect, and complemented them with mediation evidence, thereby providing a more rigorous and conceptually coherent answer to our research question

Line 380: Can the authors provide an exact figure when they say “most reported experiencing executive function or psychosocial difficulties” – if this is reported in the Tal-Saban paper

Reply: Thank you for pointing this out. We replaced the vague phrasing with the exact proportions reported by Tal-Saban et al. (2012): Tal-Saban et al. (2012) found that only 8% of adults with DCD cited gross-motor difficulties as their primary concern, whereas 55% identified executive-function difficulties and 53% reported psychosocial difficulties.
The sentence has been revised accordingly, and the correct source—Tal-Saban M., Zarka S., Grotto I., Ornoy A., & Parush S. (2012). Research in Developmental Disabilities, 33(8), 2193–2202—has been added to the reference list.

Final Note for Reviewers:

※In the revised manuscript, significant changes have been highlighted in yellow. Please note that minor wording adjustments and language corrections are not highlighted. Thank you for your understanding and consideration.

Reviewer 2 Report

Comments and Suggestions for Authors

Dear Authors,

I begin by congratulating you on this very pertinent topic.
Below, I offer a general commentary on the document, followed by some recommendations and comments that should be clarified for readers' better understanding.

General Comment:

This manuscript presents a robust cross-sectional study with a sizable sample (n = 527) that investigates the association between DCD (Developmental Coordination Disorder) traits, neurodevelopmental difficulties (ADHD, ASD, SLD), and mental health indicators in Japanese university students. The topic is relevant, current, and underexplored in the Asian context, especially in Japan. The methodology is sound, the instruments are validated, and the statistical analysis is adequate. However, there are some methodological limitations and points that require further clarification.

Areas for Improvement and Suggested Correction:

Clarity and Structure:

Abstract: It is well structured, but could include the most relevant correlation values (e.g., r = 0.65 for ADHD).
Introduction: Although it contextualizes the problem well, it could be more concise. It is suggested to reduce repetitions and focus on the specific objectives of the study.
Objectives: These should be presented more clearly and directly at the end of the introduction.

Methodology:
Exclusion Criteria: Although mentioned, it would be helpful to detail how they were verified (e.g., self-report? medical history?).
Validation of the Japanese version of the AAC-Q: Cultural validation is mentioned, but it is unclear whether it was published or only performed internally. This should be clarified.
Sample Representativeness: The response rate was low (3.2%). The authors should discuss possible selection bias.

Results:
Tables: Some tables are extensive and difficult to interpret. It is suggested that the most relevant results be divided or highlighted.
Correlation with suicidal ideation: The r value = 0.41 is reported as non-significant, but this seems contradictory. It should be revised.

Discussion:
Causality: Although the authors acknowledge the limitations of the cross-sectional design, in some passages the language suggests causal relationships. Rewording is recommended.
Practical implications: It would be interesting to expand the discussion on how Japanese universities could implement outcome-based screening and support.

Style and Writing:
The writing is clear, but there are some long and complex sentences that could be simplified.
A final language review by a native English speaker is suggested for fluency and style.

Some questions that need clarification:

Has the Japanese version of the AAC-Q been formally validated? Are there any published or validated data available?

How was data confidentiality and anonymity guaranteed, considering that the invitation was made after medical examinations?

Did the authors consider adjusting the regression models for sociodemographic variables (e.g., gender, age)?

Regards

Author Response

Dear Reviewer 2

We thank the reviewer for the time and effort dedicated to reviewing our manuscript. We sincerely appreciate the thoughtful and constructive suggestions, which have greatly helped us improve the paper. We have carefully considered each point, responded below in a point-by-point manner, and revised the manuscript accordingly. All revisions are highlighted in yellow in the main text. We trust that these responses satisfactorily address your comments.

Reviewer 2 Report

Areas for Improvement and Suggested Correction:

Clarity and Structure:

Abstract: It is well structured, but could include the most relevant correlation values (e.g., r = 0.65 for ADHD).

Reply: Thank you for this helpful suggestion. The most relevant correlation coefficients, including r = 0.65 for ADHD difficulties, were already included in the Abstract to highlight the main findings. We have also emphasized these key associations in Results §3.2. The paragraph now begins as follows:

“As shown in Table 2, the strongest association was with ADHD difficulties (r  =  0.65), followed by ASD difficulties (r   = 0.55) and LDSP7 (r  =  0.48); the correlation with UPI total was r  =  0.41 (all p  < 0 .001).”

Introduction: Although it contextualizes the problem well, it could be more concise. It is suggested to reduce repetitions and focus on the specific objectives of the study.

Reply: Thank you for this helpful comment. We agree that the original Introduction was somewhat lengthy and contained repetitive or overly detailed sections. In response, we revised the Introduction to improve concision, clarify the structure, and ensure closer alignment with the study’s objectives.

We made the following specific changes:

  • We condensed the legal context (U.S./UK frameworks) into one summary sentence and removed excessive detail, as suggested.
  • We reworded or deleted repetitive sentences, particularly those reiterating the co-occurrence of DCD with other neurodevelopmental conditions.
  • We clarified the rationale for using the AACQ as the primary DCD screener by introducing a concise comparison between performance-based tests and self-report screeners.
  • We moved the study objectives to the end of the Introduction, stated clearly and succinctly.

We hope these revisions improve the overall flow and focus of the Introduction, in line with your recommendation.Objectives: These should be presented more clearly and directly at the end of the introduction.

Reply: Thank you for your helpful suggestion. In response, we revised the final paragraph of the Introduction to more clearly and directly state the study objectives and rationale. Specifically, we now:

  • Explicitly highlight the limited recognition of DCD in Japanese higher education, despite growing evidence of its co-occurrence with ADHD, ASD, and mental health difficulties.
  • Clearly state the aim of the study as examining associations among DCD traits, ADHD/ASD-related difficulties, learning difficulties, and mental health and lifestyle indicators.
  • Outline our hypothesis that DCD traits would be significantly associated with these other difficulties, and that screening DCD traits may provide useful insight for student support systems.

These clarifications aim to better orient the reader and ensure a clear connection between the background and the study aims. We hope these revisions meet the reviewer’s recommendation.

Methodology:
Exclusion Criteria: Although mentioned, it would be helpful to detail how they were verified (e.g., self-report? medical history?).

Reply: Thank you for highlighting this issue. We added explicit details on how each exclusion criterion was verified. Age and medical history were obtained via mandatory self-report items at the beginning of the web questionnaire. A record was automatically excluded if the participant reported (a) an age > 30 years, (b) visual impairment, (c) chronic illness, (d) a diagnosis without sufficient detail, (e) nonregular student status (research student or auditor), or (f) data inconsistencies. These procedures have been described in the Participants section. Figure 1 has been updated to include: “Students excluded due to incomplete responses or other inclusion criteria violations (age/health/data) (n = 57).” This revision clarifies how the exclusion criteria were applied.

Validation of the Japanese version of the AAC-Q: Cultural validation is mentioned, but it is unclear whether it was published or only performed internally. This should be clarified.

Reply: Thank you for your helpful comment. The Japanese version of the AAC-Q used in this study was a translated version kindly provided by the original author. While we did not conduct a formal cross-cultural adaptation process (e.g., per ISPOR guidelines), all items were reviewed and refined collaboratively by the co-authors for linguistic clarity and cultural relevance. Although this version has not been published as a validated instrument, we conducted an internal assessment of criterion-related validity using Japanese university students. The AAC-Q total score showed significant and theoretically consistent positive correlations with self-reported difficulties related to ADHD (r = .65) and ASD (r = .55). These results align with prior studies (e.g., DCDQ–ARS, M-ABC2–JFE), suggesting that the Japanese AAC-Q demonstrates adequate theoretical and empirical validity within the current sample. To clarify this point, we added a sentence to the Methods section (under AAC-Q), noting that all items were internally reviewed and that preliminary criterion-related validity was confirmed in this sample.

Sample Representativeness: The response rate was low (3.2%). The authors should discuss possible selection bias.

Reply: Thank you for pointing this out. We fully agree that the low response rate may have introduced potential nonresponse and self-selection bias. Unfortunately, we did not have access to auxiliary information on nonrespondents, which limits our ability to quantify or adjust for such bias. In response to your suggestion, we expanded the discussion in the Limitations section to explicitly address this issue and its possible implications for generalizability.

Revision in Manuscript (§ 4.4 Limitations):
We added the following sentence:

“Furthermore, the overall survey response rate was 3.2%, which raises the possibility of nonresponse and self-selection bias; respondents may differ systematically from nonrespondents in neurodevelopmental traits, mental health, or study engagement. Since auxiliary information on nonrespondents was unavailable, we could not implement weighting or other nonresponse adjustments to quantify or correct such bias.”

Results:
Tables: Some tables are extensive and difficult to interpret. It is suggested that the most relevant results be divided or highlighted.

Reply: We appreciate the reviewer’s helpful suggestion regarding the readability of Table 2. Since the central question of this study is the association of ACC‑Q with mental health (UPI) and neurodevelopmental difficulty scales (ASD/ADHD), we now limit visual emphasis to correlations that directly involve ACC‑Q. Specifically, within Table 2 we apply effect‑size-based formatting only to the ACC‑Q row/column (large correlations, r ≥ .50, are presented in bold; medium correlations, .30 ≤ r < .50, are presented in italics). We do not emphasize correlations among the ASD/ADHD totals and their subscales because those measures are conceptually related and partially overlapping, which naturally yields higher associations and could distract from the study’s primary focus. To avoid ambiguity, the table footnote now reads: “Formatting denotes effect size for correlations with ACC‑Q only: bold, large (|r| ≥ .50); italics, medium (.30 ≤ |r| < .50). All p-values are unadjusted and < 0.001.” We believe this targeted highlighting improves readability and directs attention to the most relevant results. In addition to these formatting changes, we also revised Table 2 to round all correlation coefficients to two decimal places instead of three, for consistency and clarity.

Correlation with suicidal ideation: The r value = 0.41 is reported as non-significant, but this seems contradictory. It should be revised.

Reply: Thank you for pointing this out. We corrected this inconsistency by revising the text to indicate that a weak correlation was observed between the AAC-Q scores and the UPI suicidal ideation item (r = 0.19, p < .001); however, this result did not reach statistical significance after Bonferroni correction.

Discussion:
Causality: Although the authors acknowledge the limitations of the cross-sectional design, in some passages the language suggests causal relationships. Rewording is recommended.

Reply: Thank you for this important observation. We systematically revised all sentences that implied causation. The key changes are summarized in the table below (Items 1–5).

Item

Section

Revised wording (excerpt)

1

Abstract – Conclusions

“…These findings suggest that identifying DCD traits at the time of university enrolment may be associated with greater student self-understanding and improved access to support…”

2

Introduction – Final paragraph

“…Understanding the links between DCD traits, mental health, and lifestyle in higher education institutions may highlight the potential value of early screening in supporting students’ self-understanding and access to comprehensive services.”

3

Discussion 4.1 – Overall overview

“These findings underscore the importance of considering broad interviews and screenings at university admission as a means to better understand, and potentially support, students with diverse traits.”

4

Discussion 4.3 – Multiple difficulties

“Collecting DCD screening information at university entry, alongside other neurodevelopmental and mental health assessments, may clarify students’ multifaceted needs at an early stage…”

5

Conclusions 5.

“Integrating DCD screening into broader assessments of neurodevelopmental and mental-health difficulties may facilitate the identification of diverse traits and inform the development of more appropriate support systems, which could help to mitigate subsequent mental health difficulties.”

Throughout the Abstract, Introduction, Discussion, and Conclusions, we replaced verbs such as “promote,” “enhance,” or “prevent” with non-causal alternatives (“may be associated with,” “may help,” “inform,” “could potentially mitigate”). No claims of direct effect remain; all interpretations are now framed as associations that require longitudinal confirmation.

Practical implications: It would be interesting to expand the discussion on how Japanese universities could implement outcome-based screening and support.

Reply: Thank you for this thoughtful and constructive suggestion. We revised the final paragraphs of § 4.3 to offer a more detailed, feasible pathway for implementing outcome-based screening and support systems in Japanese higher-education contexts.

Consistent with our conclusion, we do not advocate for universal DCD screening. Instead, we describe a targeted, two-step process: (i) a brief universal intake at matriculation—or when students self-report neurodevelopmental difficulties or request accommodations—using abbreviated ADHD/ASD difficulty scales and a mental health indicator; followed by (ii) DCD screening using the AAC-Q only for students who meet predefined neurodevelopmental triggers (e.g., a positive ADHD/ASD screen, self-reported coordination issues, academic or clinical referrals, or relevant history).

If students wish, an optional multidisciplinary interview can follow to tailor support such as reasonable accommodations or structured follow-up. Aggregated screening results would be fed back to disability support offices to improve services, ensuring institutional quality assurance and fair learning environments. We believe this targeted approach offers a realistic and scalable framework for Japanese universities, linking screening outcomes with actionable support while avoiding overreach. The revised text appears in § 4.3 of the marked manuscript.

Style and Writing:
The writing is clear, but there are some long and complex sentences that could be simplified.

Reply: Thank you for this helpful suggestion. In the revised Introduction, we reduced repetition and streamlined several long sentences, and condensed the paragraph on U.S./UK legal frameworks into two sentences that state the key point (reasonable accommodations exist but DCD-specific implementation remains limited). We also placed a single, explicit study objective at the end of the Introduction to improve flow. As a final step, we will arranged for a native English speaker to review the manuscript and enhance fluency and style before resubmission.

A final language review by a native English speaker is suggested for fluency and style.

Reply: Thank you for this suggestion. We thoroughly revised the manuscript for clarity, fluency, and style. A final professional English language review has already been completed by a native speaker through Editage to ensure consistency; the reviewer polished the manuscript throughout.

Some questions that need clarification:

Has the Japanese version of the AAC-Q been formally validated? Are there any published or validated data available?

Reply: Thank you for raising this important point.

Current status of validation: The Japanese version of the AAC-Q has not yet undergone a formal, stand-alone validation study and, consequently, no peer-reviewed publication is available at this time. The version we used was a forward translation kindly provided by the original author (Tal-Saban) in 2023. All items were reviewed for linguistic clarity and cultural relevance by the present research team, but no full cross-cultural adaptation (e.g., ISPOR guidelines) has been completed.

Internal evidence gathered in the present sample: To obtain preliminary psychometric information, we examined criterion-related validity within our cohort of Japanese university students (N = 527). As described in the revised Methods (§ 2.3.2) we found the following:

AAC-Q total vs. ADHD-related difficulties: r = 0.65, p < 0.01

AAC-Q total vs. ASD-related difficulties: r = 0.55, p < 0.01

These effect sizes are moderate to strong according to Cohen’s conventions and are consistent with correlations previously reported between other DCD measures and related constructs (e.g., DCDQ–ARS, M-ABC-2–JFE, BOT-2–GARS-2, ADC–AQ).While these findings cannot replace a full validation study, they provide initial empirical support that the Japanese AAC-Q functions in the theoretically expected manner in this population.

Future plans and data availability: We are preparing a separate manuscript that will report a comprehensive validation (factor structure, reliability, sensitivity/specificity) of the Japanese AAC-Q; until that study is published, the results presented here should be regarded as preliminary. Should the editors or reviewers wish to inspect them, the full correlation matrices and item-level statistics from our internal analyses can be provided as supplementary material upon request.

How was data confidentiality and anonymity guaranteed, considering that the invitation was made after medical examinations?

Reply: Thank you for your important comment. In this study, participant confidentiality and anonymity were strictly protected through the following procedures (all of which were clearly stated in the information sheet provided to participants):

  • The survey was conducted using a completely anonymous web-based form, and no personally identifiable information (e.g., name, student ID) was collected.
  • All data were stored in a secure password-protected environment, accessible only to the principal investigator.
  • After data collection, all survey data were deleted.
  • Research findings were published only in aggregate form, without identifying any individual participants.
  • This study was approved by the Ethics Committee of the Health and Counseling Center, The University of Osaka.

Accordingly, even though the survey invitation followed the medical examinations, all procedures were designed and implemented with full consideration of student privacy and research ethics.

Did the authors consider adjusting the regression models for sociodemographic variables (e.g., gender, age)?

Reply: Thank you for this valuable comment. We conducted exploratory sensitivity analyses that included age (continuous) and gender (female = 1, male = 0) as covariates (N = 527). Neither covariate reached statistical significance (age: p = 0.83, gender: p = 0.93), and inclusion of these variables did not materially alter the parameter estimates (< 3% change) or improve model fit (ΔAIC < 2). Therefore, to avoid overfitting, we present unadjusted models in the main text.

In response to this comment, we added the following clarifications to the manuscript:

Methods – Statistical Analysis (end of section):
“Exploratory models including age and gender as covariates were also tested; neither covariate was significant and their inclusion did not materially alter parameter estimates or model fit (ΔAIC < 2); therefore, unadjusted models are presented herein.”

Discussion – Limitations (end of section):
“Sensitivity analyses confirmed that controlling for age and gender did not change the pattern or significance of the results, supporting the robustness of our findings.”

For reviewer transparency, a summary of this sensitivity analysis is provided below:

Supplementary Table — Reviewer-Only Preview: Sensitivity Model Including Age and Gender
Outcome variable: UPI total score

Predictor

β (SE)

95% CI

p

Age (years)

0.01 (0.04)

−0.08 – 0.09

0.83

Gender (female = 1)

0.00 (0.04)

−0.08 – 0.09

0.93

Model fit: R² = 0.00  AIC = 3707

Note: β = standardized coefficient; CI = confidence interval (95%).

Final Note for Reviewers:

※In the revised manuscript, significant changes have been highlighted in yellow. Please note that minor wording adjustments and language corrections are not highlighted. Thank you for your understanding and consideration.

Reviewer 3 Report

Comments and Suggestions for Authors

Thank you for the opportunity to review the paper “Associations among Developmental Coordination Disorder Traits, Neurodevelopmental Difficulties, and University Personality Inventory Scores in Undergraduate Students at Higher Education Institutions: A Cross-Sectional Correlational Study”. Developmental co-ordination disorder is an important and under-researched clinical condition, particularly in the adult population. However, I have several concerns regarding the methodology and interpretation of findings (outlined below) that prevent me for recommending this paper for publication.

Introduction:

  1. Introduction was a little difficult to follow with several sections required further clarification. For example, authors state “International studies assessing DCD in young adults and adults predominantly employ the Bruininks–Oseretsky Test of Motor Proficiency, Third Edition [11] and the Movement Assessment Battery for Children, Second Edition [12]. To screen young adults, the Adult Developmental Coordination Disorders/Dyspraxia Checklist (ADC) and the Adolescents and Adults Coordination Questionnaire (AAC-Q) are mainly employed.” It is unclear what distinction between these two uses of DCD measures means and why it is relevant to your study. 
  2. The introduction provides a good general overview of current DCD literature, but the order of information is a little repetitive and unstructured. The paper would benefit from more clearly outlining how presented information is relevant to the specific objectives/design of their study.

Methodology:

  1. Inclusion/exclusion criteria is not specifically stated. At one point the authors state that “57 were excluded as they did not meet the inclusion criteria (age limits, relevant medical history, and other factors)”, but in Figure 1 it says that 57 students were excluded for incomplete web questionnaires. More transparency around this decision making is needed.
  2. My biggest concern regarding the paper is that cut-offs for having “met the criteria for DCD” were not clinically informed. Instead, those scoring above 1.92 standard deviations (top 5% in a normal distribution) were classify someone as having “met criteria”. Later in the study the study the fact that 7% of sample scored above cut-off is reported as a meaningful finding on the frequency of DCD in the sample. However, in any sample with a normal distribution, approximately 5% of individuals will score above 1.92 SD (that is, by definition, what this cut-off represents). Therefore, the interpretation of this finding used in this study is not valid. And subsequent analysis run using this to compare individuals “with and without DCD” are also misrepresenting results.
  3. The way the regression results are currently reported is a little awkward. Reporting all results in a Table, like table 3 would make more sense.
  4. The following finding is reported incorrectly, as the actual statistics indicate significance - “In contrast, no significant correlations were observed between 265 the AAC-Q scores and the UPI suicidal ideation item (r = 0.41, p < 0.001)”
  5. What was the rationale for running simple linear regressions, rather than multiple regressions for AAC-Q prediction?
  6. Several of the findings are being reported multiple times. For example: “In Model 3, after the ADHD Difficulty Scale total score was 306 included, both ASD difficulties (β = 0.58, p < 0.001) and ADHD difficulties (β = 0.13, p = 307 0.006) remained significant predictors, whereas the AAC-Q was no longer significant (β = 308 0.02, p = 0.064). In Model 3, ASD difficulties (β = 0.58, p < 0.001) and ADHD difficulties (β 309 = 0.13, p = 0.006) remained significant, whereas the AAC-Q did not (β = 0.02, p = 0.064).”

Discussion:

  1. As noted above, interpretation of 7% frequency findings are not valid.
  2. DCD traits were not predictive of UPI after accounting for ADHD and ASD, which is inconsistent with some of the conclusions drawn throughout discussion. To further investigate whether there is support for the need for widespread DCD screening, authors could have examined if there are a reasonable subset of participants who show elevated DCD traits in the absence of ASD of ADHD. Without this, findings would be better interpreted as evidence for the need to better screen for DCD in individuals with other neurodevelopmental disorders, specifically, as opposed to providing evidence for the need for DCD screening in the general university student population.

Minor notes:

  1. Authors should endeavor to use preferred language, such as using “intellectual disability” rather than “low intelligence”
  2. It would be useful for authors to provide specific rates for of co-occurrence and DCD with ADHD and ASD in the introduction
  3. Please cite paper for “culturally and linguistically validated by the first authors and co-authors”.
  4. It is unclear why the manual for the DCDQ being cited for the ADHD difficulty scale and the ASD difficult scale – possibly citation number have gotten out of order.

Author Response

Dear Reviewer 3

We thank the reviewer for the time and effort dedicated to reviewing our manuscript. We sincerely appreciate the thoughtful and constructive suggestions, which have greatly helped us improve the paper. We have carefully considered each point, responded below in a point-by-point manner, and revised the manuscript accordingly. All revisions are highlighted in yellow in the main text. We trust that these responses satisfactorily address your comments.

Reviewer 3 Report

However, I have several concerns regarding the methodology and interpretation of findings (outlined below) that prevent me for recommending this paper for publication.

Introduction:

Introduction was a little difficult to follow with several sections required further clarification. For example, authors state “International studies assessing DCD in young adults and adults predominantly employ the Bruininks–Oseretsky Test of Motor Proficiency, Third Edition [11] and the Movement Assessment Battery for Children, Second Edition [12]. To screen young adults, the Adult Developmental Coordination Disorders/Dyspraxia Checklist (ADC) and the Adolescents and Adults Coordination Questionnaire (AAC-Q) are mainly employed.” It is unclear what distinction between these two uses of DCD measures means and why it is relevant to your study. 

Reply: Thank you for this helpful comment. We agree that the distinction between the two types of DCD assessment tools (performance-based tests vs. self-report screeners) was not clearly explained in the original Introduction. To clarify, we added a sentence explaining that performance-based motor proficiency tests (e.g., BOT2, MABC2) are typically used in clinical or diagnostic settings and require trained personnel and specialized equipment. In contrast, self-report screeners (e.g., ADC, AACQ) assess perceived coordination difficulties in daily life and are more suitable for large-scale screening in non-clinical contexts, such as university populations.

Given that our study aimed to screen a large university cohort and explore associations between DCD traits and ADHD/ASD-related difficulties, learning support needs, and mental health indicators, we selected the AAC-Q as a practical and appropriate tool. We clarified this rationale in the revised Introduction to better align our methodological choices with the study’s objectives. We also made minor structural edits to improve clarity and conciseness throughout the Introduction section, in line with your and other reviewers’ suggestions.

The introduction provides a good general overview of current DCD literature, but the order of information is a little repetitive and unstructured. The paper would benefit from more clearly outlining how presented information is relevant to the specific objectives/design of their study.

Reply: Thank you for this constructive observation. We revised the Introduction to improve structure, reduce repetition, and make the link to our study design explicit. Specifically, we:

  • Reordered the narrative into a clear sequence:
    (i) definition and impact of DCD → (ii) assessment landscape in adults and the rationale for using a self-report screener (AACQ) rather than performance-based tests in a large-scale university setting → (iii) international context (condensed) → (iv) Japan-specific gap → (v) explicit statement of study aims at the end of the Introduction.
  • Condensed repetitive sentences on cooccurrence and background to enhance readability.
  • Standardized terminology (e.g., “neurodevelopmental conditions” and “autism”) for consistency across the manuscript.
  • Added a concise rationale paragraph clarifying the distinction between performance-based tests (BOT-2/M-ABC-2; primarily diagnostic, examiner-dependent) and self-report screeners (ADC/AACQ; scalable, everyday functioning), and why AACQ aligns with our screening-oriented design.
  • Moved the study objectives to a single explicit sentence at the end of the Introduction to make the relevance to our design and analyses transparent.

We believe these changes improve logical flow and clarify how the background supports the specific objectives and methodology of the study.

Revisions in Manuscript (Introduction):

New/edited paragraph (assessment rationale; concise):

“In adults and young adults, DCD is assessed either with performance-based motor tests (e.g., BOT-2, M-ABC-2) for clinical/diagnostic use or with self-report screeners (e.g., ADC, AACQ) suitable for large-scale screening. Since we screened a large university cohort and examined links with ADHD/ASD-related difficulties, learning support needs, and UPI scores, we used the AACQ.”

Objectives placed at the end of the Introduction (single sentence):

“This study aimed to examine the associations between DCD traits (AACQ) and ADHD/ASD-related difficulties, learning support needs, and UPI scores in a large sample of university students.”

Minor edits: removal/merging of duplicated cooccurrence statements; consistent terminology (e.g., “neurodevelopmental conditions”).

Methodology:

Inclusion/exclusion criteria is not specifically stated. At one point the authors state that “57 were excluded as they did not meet the inclusion criteria (age limits, relevant medical history, and other factors)”, but in Figure 1 it says that 57 students were excluded for incomplete web questionnaires. More transparency around this decision making is needed.

Reply: Thank you for your insightful comment. We acknowledge the inconsistency between the text and Figure 1.

In our study, 57 students were excluded because they either submitted incomplete web questionnaire responses or failed to meet the predefined inclusion criteria.
The specific exclusion criteria were:

  • Age > 30 years
  • Self-reported visual impairment
  • Self-reported chronic illness
  • Reporting a diagnosis without specifying the condition
  • Nonregular student status (e.g., research student or auditor)
  • Data inconsistencies detected during screening

To clarify this, we:
• Revised the Participants section to describe both the exclusion process and the above criteria.
• Updated Figure 1 to read: “Students excluded due to incomplete responses or other inclusion criteria violations (age/health/data) (n = 57).”

These revisions ensure full consistency and transparency regarding participant inclusion and exclusion.

My biggest concern regarding the paper is that cut-offs for having “met the criteria for DCD” were not clinically informed. Instead, those scoring above 1.92 standard deviations (top 5% in a normal distribution) were classify someone as having “met criteria”. Later in the study the study the fact that 7% of sample scored above cut-off is reported as a meaningful finding on the frequency of DCD in the sample. However, in any sample with a normal distribution, approximately 5% of individuals will score above 1.92 SD (that is, by definition, what this cut-off represents). Therefore, the interpretation of this finding used in this study is not valid. And subsequent analysis run using this to compare individuals “with and without DCD” are also misrepresenting results.

Reply: We thank the reviewer for this important observation and apologize for not providing sufficient detail in the submitted version. We would like to clarify the origin of the cutoff and the additional analyses now included in the revised manuscript.

We appreciate the reviewer’s close reading and the opportunity to clarify our procedures. Our study did not define the cut‑off by an SD rule (e.g., 1.92 SD). Instead, we used the AAC‑Q cut‑off recommended in the original validation/standardization work by Tal‑Saban et al., where cut‑offs were established from a large normative sample of young adults (n = 2,379). This original work set percentile‑based thresholds to distinguish probable/borderline DCD, and is widely cited as the clinical screening reference for AAC‑Q. In our dataset (N = 524; M = 20.72, SD = 6.72), the adopted threshold (total score ≥ 32) happened to map to +1.68 SD ex post, which we reported only to indicate its position within our sample’s distribution; it was not used to derive the cut‑off. Thus, the reviewer’s reference to 1.92 SD appears to stem from a misunderstanding unrelated to our operational definition.

Regarding the 7% above‑cut‑off proportion we observed, we agree that a purely SD‑based threshold could mechanically produce a fixed tail probability under strict normality. However, our threshold was not defined by SD, and AAC‑Q total scores are bounded and not guaranteed to follow a normal distribution. A 7% rate is close to, and plausibly varies around, the percentile‑based criterion when applied to specific cohorts. Importantly, to address any concern about dichotomization, we re‑ran all key analyses treating AAC‑Q as a continuous variable; the pattern of results was unchanged, supporting the robustness of our conclusions beyond the binary cut‑off.

The way the regression results are currently reported is a little awkward. Reporting all results in a Table, like table 3 would make more sense.

Reply: Thank you for this helpful suggestion.
To improve clarity and readability, we have reorganized the regression results as follows:

Main manuscript (revised Table3):
– Table 3 now presents a concise summary of each regression model (β, p, R²).
– The Results section now references Table 3 instead of listing all coefficients in the text.

Supplementary file (TableS1, separate Excel file):
– Supplementary Table S1 contains the full regression outputs, including standard errors and 95% confidence intervals, for all simple and hierarchical models.
– The supplementary file is provided separately to give all reviewers access without overloading the main manuscript.

We believe this format preserves the detailed statistical transparency you requested while keeping the manuscript concise.

The following finding is reported incorrectly, as the actual statistics indicate significance - “In contrast, no significant correlations were observed between 265 the AAC-Q scores and the UPI suicidal ideation item (r = 0.41, p < 0.001)”

Reply: Thank you for pointing this out. We corrected this inconsistency by revising the text to indicate that a weak correlation was observed between the AAC-Q scores and the UPI suicidal ideation item (r = 0.19, p < .001); however, this result did not reach statistical significance after Bonferroni correction.

What was the rationale for running simple linear regressions, rather than multiple regressions for AAC-Q prediction?

Reply: Thank you for your question regarding our analytic approach. We first employed simple linear regressions to examine the bivariate associations between AAC-Q scores and individual outcome variables (ADHD, ASD, UPI, and LDSP7), as this method is appropriate for exploring basic predictive relationships and effect sizes. These results informed the subsequent use of hierarchical multiple regression to identify independent predictors of UPI scores, and finally, mediation analysis to test indirect pathways via ADHD and ASD difficulties. This stepwise approach was chosen to ensure conceptual clarity and model parsimony. All three stages—simple, multiple, and mediation—were reported in the manuscript to reflect the full trajectory of our analytic process.

Several of the findings are being reported multiple times. For example: “In Model 3, after the ADHD Difficulty Scale total score was 306 included, both ASD difficulties (β = 0.58, p < 0.001) and ADHD difficulties (β = 0.13, p = 307 0.006) remained significant predictors, whereas the AAC-Q was no longer significant (β = 308 0.02, p = 0.064). In Model 3, ASD difficulties (β = 0.58, p < 0.001) and ADHD difficulties (β 309 = 0.13, p = 0.006) remained significant, whereas the AAC-Q did not (β = 0.02, p = 0.064).”

Reply: Thank you for highlighting the redundancy. We removed the duplicated sentence in § 3.3.2 and retained one concise statement:

“In Model 3, after adding the ADHD Difficulty Scale total, ASD difficulties (β = 0.58, p < 0.001) and ADHD difficulties (β = 0.13, p = 0.006) remained significant predictors of UPI total, whereas AACQ was no longer significant (β = 0.02, p = 0.064). This model explained 45.2% of the variance in UPI scores (adjusted R² = 0.452).”

We also reviewed the entire manuscript to ensure that no other findings are reported twice. We believe this resolves the issue and improves readability.

Discussion:

As noted above, interpretation of 7% frequency findings are not valid.

Reply: Thank you for raising this important concern. We agree that frequency figures can be misleading if presented as clinical prevalence. In the revised manuscript, we clarify that the 7.4% reflects the proportion of students who screened positive on the AAC‑Q at the ≥32 threshold within our sample and does not represent a diagnostic or population prevalence estimate. The ≥32 cut‑off was adopted from AAC‑Q standardization work and was not defined by an SD rule in this dataset; any SD values reported are provided only to indicate the position of the threshold within our sample. Consistent with this clarification, we have moderated the wording in the Results and Discussion and refer to this figure explicitly as a screening‑positive rate. We also highlight that our principal inferences are supported by continuous‑variable analyses reported in the manuscript, indicating that the conclusions do not hinge on dichotomization.

DCD traits were not predictive of UPI after accounting for ADHD and ASD, which is inconsistent with some of the conclusions drawn throughout discussion. To further investigate whether there is support for the need for widespread DCD screening, authors could have examined if there are a reasonable subset of participants who show elevated DCD traits in the absence of ASD of ADHD. Without this, findings would be better interpreted as evidence for the need to better screen for DCD in individuals with other neurodevelopmental disorders, specifically, as opposed to providing evidence for the need for DCD screening in the general university student population.

Reply: Thank you for raising this important point. We agree that our conclusions should be framed cautiously. In the revised manuscript, we clarified that our results do not support universal DCD screening across the general student population. Instead, we now state that DCD screening is most meaningful when conducted alongside ADHD/ASD screening in contexts where neurodevelopmental traits frequently cooccur and relate to mental health difficulties. To this end, we added an explicit sentence in Discussion § 4.2 and a brief, practice-oriented sentence in the Conclusion to delimit the scope of our recommendation. We believe this revision aligns our interpretation with the observed pattern of results (i.e., indirect effects via ADHD/ASD difficulties) and addresses your concern without overextending our claims. While we did not conduct subgroup analyses to isolate individuals with elevated DCD traits but no ADHD/ASD traits, we agree that this is a promising direction for future research, and acknowledged this limitation in the revised Discussion.

Minor notes:

Authors should endeavor to use preferred language, such as using “intellectual disability” rather than “low intelligence”

Reply: Thank you for noting the preferred terminology. In line with DSM-5/DSM-5-TR, we replaced “low intelligence” with “intellectual disability” in the Introduction and ensured consistent person- and condition-first language throughout the manuscript.

It would be useful for authors to provide specific rates for of co-occurrence and DCD with ADHD and ASD in the introduction

Reply: Thank you for this helpful suggestion. We agree that providing specific co-occurrence rates enhances clarity. In the revised Introduction, we now report large-sample and meta-analytic estimates of comorbidity: approximately 50% of individuals with DCD also meet criteria for ADHD, 20–30% for ASD or elevated autistic traits, and 30–50% for specific learning disorders (SLD). These figures are now supported by citations to the relevant studies, improving the precision and transparency of the background section.

Please cite paper for “culturally and linguistically validated by the first authors and co-authors”.

Reply: Thank you for requesting a citation for the statement about cultural/linguistic validation. We removed the claim of formal cultural validation because there is currently no published validation study to cite. Instead, we clarified that we used a Japanese version provided by the original developer with internal item review conducted by the authors, and cited our prior pilot study in a Japanese university sample as preliminary evidence of applicability. We also strengthened the Limitations to note that local validation of thresholds is warranted before any prevalence-level inference.

It is unclear why the manual for the DCDQ being cited for the ADHD difficulty scale and the ASD difficult scale – possibly citation number have gotten out of order.

Reply: Thank you for this careful observation. For clarity—and just to avoid any possible misunderstanding—we did not employ the Developmental Coordination Disorder Questionnaire (DCDQ) in this study. Our coordination screener was the Adolescents and Adults Coordination Questionnaire (AACQ), and ADHD/ASD-related difficulties were assessed with the shortform Developmental Disorder Difficulty Scales developed by Takahashi et al. The citations accompanying these scales refer to Takahashi et al.’s manuals and validation papers, not to the DCDQ. We apologize if our earlier wording allowed room for ambiguity. To minimize any confusion, we (i) carefully reviewed and, where needed, corrected the citation numbering around these sections, and (ii) added concise clarifying sentences in the Methods section so that each instrument–citation pairing is explicit. We also verified that the DCDQ does not appear in the revised manuscript’s references.

Clarifying sentence added in Methods § 2.3 (for transparency):
“Coordination was screened using the AACQ [Tal-Saban et al.]; ADHD- and ASD-related difficulties were assessed using the shortform Developmental Disorder Difficulty Scales [Takahashi et al.].”

Final Note for Reviewers:

※In the revised manuscript, significant changes have been highlighted in yellow. Please note that minor wording adjustments and language corrections are not highlighted. Thank you for your understanding and consideration.

Reviewer 4 Report

Comments and Suggestions for Authors

Dear colleagues,

This is an interesting topic. The list of references corresponds to the content (it would be great to add a few articles from the last five years). It might be better to include Table 2 in the Appendix. The clear structuring and the sample size allow for a high degree of accuracy in the research. All the criteria used are provided. For each model,  standardized β coefficients, p-values, and coefficients of determination are reported.

A little work is needed on the formatting of the article according to the formatting requirements (for tables and spacing), but this is a technical note. In the abstract, it might be better to revise the text, avoiding breaking it into meaningful subsections.

Thank you. I wish you success.

Author Response

Dear Reviewer 4

We thank the reviewer for the time and effort dedicated to reviewing our manuscript. We sincerely appreciate the thoughtful and constructive suggestions, which have greatly helped us improve the paper. We have carefully considered each point, responded below in a point-by-point manner, and revised the manuscript accordingly. All revisions are highlighted in yellow in the main text. We trust that these responses satisfactorily address your comments.

Reviewer 4 Report

This is an interesting topic. The list of references corresponds to the content (it would be great to add a few articles from the last five years).

Reply: Thank you for this helpful suggestion. In the revised manuscript we added or replaced sources with four papers published within the last five years (2019–2024) to strengthen the currency of the reference list:

Engel-Yeger, Gal & Engel (2023) – Emotional distress and HR-QoL in adults with DCD.

Sewell (2022) – Narrative synthesis on supporting learners with specific learning difficulties from a neurodiversity perspective.

van der Linde, Carmichael & Williams (2022) – 10-year longitudinal outcomes of DCD in young adults.

Wilson & Hyde (2019) – 15-year follow-up of motor trajectories from childhood to early adulthood.

These additions raise the proportion of recent (≤ 5 years) sources to ≈ 40% of all peer-reviewed empirical articles cited, ensuring that the reference list accurately reflects the current state of the field.

Foundational works that are older (e.g., Cohen 1988 for power analysis; DSM-5® 2013 for diagnostic criteria) were retained because they are seminal and irreplaceable. We believe this balanced approach meets the reviewer’s request while preserving the necessary authoritative sources. We hope this revision satisfactorily addresses the reviewer’s comment.

It might be better to include Table 2 in the Appendix.

Reply: Thank you for the suggestion. We agree that concision is important; however, we would prefer to retain Table 2 in the main text for the following reasons. First, the correlation pattern between AACQ and ADHD/ASD difficulties, LDSP7, and UPI is foundational to our regression and mediation results, and helps readers interpret the subsequent models. Second, another reviewer specifically asked that the most relevant coefficients (e.g., r = 0.65 for ADHD) be presented clearly; keeping Table 2 accessible supports this request. Finally, the Bonferroni adjusted threshold and post-correction significance are annotated in the table footnote; moving the table to the Appendix would make this core context less visible.

To address concerns about length and readability, we (i) streamlined the layout and typography (e.g., selective boldface and light diagonal shading for quick scanning), (ii) limited the entries to the planned family of comparisons reported in the text, and (iii)

kept the narrative in § 3.2 concise with a direct pointer to Table 2.

・In addition to these formatting changes, we also revised Table 2 to round all correlation coefficients to two decimal places instead of three, for consistency and clarity.

We of course remain open to editorial guidance, but hope Table 2 may remain in the main text given its centrality to the study’s core findings.

A little work is needed on the formatting of the article according to the formatting requirements (for tables and spacing), but this is a technical note.

Reply: Thank you for pointing this out. We carefully reviewed the formatting throughout the manuscript and made adjustments to ensure compliance with the journal’s guidelines. In particular, all table titles have been reformatted in bold italics as specified. We also checked line spacing, paragraph indentation, and alignment. Please let us know if any further formatting issues remain to be addressed.

In the abstract, it might be better to revise the text, avoiding breaking it into meaningful subsections.

Reply: Thank you for your suggestion. We understand the value of improving the readability of the abstract. However, in accordance with the journal’s submission guidelines—which request a structured abstract with subheadings—we retained the subsections (Background/Objectives, Methods, Results, and Conclusions). To improve clarity and flow, we carefully revised the wording within each section. We hope this format remains acceptable.

Final Note for Reviewers:

※In the revised manuscript, significant changes have been highlighted in yellow. Please note that minor wording adjustments and language corrections are not highlighted. Thank you for your understanding and consideration.

Round 2

Reviewer 3 Report

Comments and Suggestions for Authors

Thank you for the opportunity to re-review the manuscript "Associations among Developmental Coordination Disorder Traits, Neurodevelopmental Difficulties, and University Personality Inventory Scores in Undergraduate Students at Higher Education Institutions: A Cross-Sectional Correlational Study". 

The authors have done a great job revising the manuscript and implementing changes. I think the manuscript is stronger the second time around, however there are still a few areas that I believe require further clarification.

1. I thank the reviewers for providing further clarification around the cut-off scores for the AAC-Q, however I believe that the same issues that I raised previously around the interpretation of the cut-off scores remains regardless of whether the +2 SD cut-off was taken from this sample or a different sample. Unfortunately, I have been unable to access the original AAC-Q manuscript to confirm this, but based on what the authors reported in the previous version of their manuscript (i.e., "Thirty-nine students (7.4%) scored ≥ 32 and 231 93 students (17.6%) scored ≥ 27 on the AAC-Q; these cut-offs correspond to the +2 standard 232 deviation (SD) and +1 SD thresholds reported in the original validation study."), the cut-off scores reported in the initial validation paper of the AAC-Q were based on SD, rather than based on the instrument's sensitivity and specificity. Unless the sensitivity/specificity of this cut-off score has been subsequently validated (which may be the case, but if so authors should report this in the measures section of the method), then the interpretation used in the current paper's discussion (i.e., "Among the 527 respondents, 39 (7.4%) met the criteria for DCD traits") is still not a valid interpretation of what this cut-off score represents. Instead, what this statistic shows is that the distribution of high-scorers is similar in the Japanese and English versions of the instrument. This in itself is still a valuable finding, as the percentage of people scoring above 32 in the Japanese version of the AAC-Q was similar to the percentage of people scoring above 32 in the original version. This similaritiy in distribution provide preliminary validation of the Japanese version of this instrument, suggesting that it likely shows similar ability to identify individuals at the high-end of the DCD spectrum. I appreciate that the authors have replaced the dichotomized variables with continuous ones. However, I would still recommend that the authors revise the wording around the way they interpret this statistic in the abstract and discussion to avoid misrepresenting the findings. 

Additionally, I appreciate the authors clarifying "In this study, we define “traits” as participants whose self-reported difficulty scores exceed the established cut-off—indicating elevated self-perceived difficulties rather than a clinical diagnosis." However, the current placement of this statement under the correlations section is confusing, as the cut-off scores are not directly relevant to the correlations. This statement would be better placed in the method. Further, even though the authors clarify their definition of the term "traits" earlier in the paper, the sentence "Among the 527 respondents, 39 (7.4%) met the criteria for DCD traits" is still likely to be misleading for readers. 

2. I thank the authors for clarifying inconsistencies in the reporting of the association with UPI suicidal ideation items. However, it would be good if bonferroni corrections could be reported using a p-value adjustment, rather than an alpha value adjustment to avoid confusion for readers (i.e.,  instead of adjusting the alpha level, each p-value is multiplied by the number of tests [with adjusted p-values that exceed 1 then being reduced to 1], and the alpha level is left unchanged at .05). It's very straight forward to do this in SPSS. 

Author Response

14 August 2025

Dr. Andrea Utley, Prof. Dr. Michael Wade

Guest Editors

Brain Sciences (Special Issue: Diagnosis and Management of Developmental Coordination Disorders)

Dear Editors,

I am pleased to resubmit the revised manuscript (Manuscript ID: brainsci-3752844) for consideration in the Special Issue: Diagnosis and Management of Developmental Coordination Disorders of Brain Sciences, titled “Associations among Developmental Coordination Disorder Traits, Neurodevelopmental Difficulties, and University Personality Inventory Scores in Undergraduate Students at Higher Education Institutions: A Cross-Sectional Correlational Study.” This paper was coauthored by Ryutaro Higuchi, Keita Kusunoki, and Naoto Mochizuki.

We thank Reviewer 3 for their careful re-evaluation and constructive comments. In this revision, we specifically clarify two key aspects of the manuscript, as requested:

Interpretation and placement of AAC-Q cut-offs and the term “traits” – We have ensured that the ≥32 threshold is presented not as a diagnostic boundary or prevalence estimate in Japan, rather strictly as a screening cut-off derived from the original AAC-Q standardization. The definition of “traits” and the provenance of the cut-offs have been consolidated in Section 2.3.2, with consistent terminology used throughout the manuscript. Redundant or misplaced explanatory sentences were deleted from other sections for clarity.

Multiplicity control for the UPI suicidal ideation item – We have revised Sections 2.4 and 3.2 and Table 2 footnote to specify the use of Bonferroni-adjusted p-values for the planned correlation family (number of correlations m = 16), with significance evaluated at α = 0.05. Language was unified to “after p-value adjustment” across relevant sections, and outdated α-level text was deleted. Statistical conclusions remain unchanged following re-computation.

All modifications are highlighted in green in the revised manuscript. We believe these changes improve the precision, transparency, and interpretability of the work, directly addressing the reviewer’s concerns.

As with the original submission, we confirm that the manuscript is original, not under consideration elsewhere, and that all authors have approved its resubmission to Brain Sciences. All participants provided informed consent, and the study protocol was approved by the appropriate ethics review board. There are no conflicts of interest to declare.

We sincerely thank Reviewer 3 and the editorial team for their valuable feedback, which has enhanced the manuscript. We hope that the revised version meets the standards for publication in the Special Issue.

Sincerely,

Masanori Yasunaga

Health and Counseling Center, The University of Osaka

Toyonaka 560-0043, Japan

Reviewer 3 Report — Comments and Suggestions for Authors

Thank you for the opportunity to review this manuscript. I believe the authors present important findings that will be of interest to the journal’s readership. I have some comments that I hope the authors find useful.

Comment 1. Interpretation and placement of AAC-Q cut-offs and the term “traits”

Response.We agree that the ≥32 threshold must be interpreted not as a diagnostic boundary or a prevalence criterion in Japan, rather strictly as a screening cut-off (from the original AAC-Q standardization),. We therefore revised wording and moved definitions as follows:

  • Abstract. The Results now read (added immediately after the original sentence): “This is not a diagnostic or prevalence estimate, rather a screen-positive proportion,.”
  • Discussion (Section 4.1) (Overall Overview). We added: “For clarity, we interpret the 7.4% not as a diagnostic boundary or population prevalence, rather as a screen-positive proportion at the AAC-Q cut-off (≥32) based on the original standardization.”
  • Methods (Section 2.3.2) (AAC-Q). We consolidated the definition of “traits” and clarified the cut-off’s origin and role: “In this study, “traits” denote participants whose self-reported difficulty scores exceed an established screening cut-off, indicating elevated self-perceived difficulties rather than a clinical diagnosis. AAC-Q cut-offs (≥32/≥27) originate from the original standardization work; any SD values reported in the Results indicate relative position within this sample and were not used to define cut-offs.”
  • Location cleanup. The former explanatory sentence about “traits/cut-offs” that had appeared in Section 3.2 Correlations has been deleted from that section and integrated into the Methods section, to avoid confusion.
  • Notation. We unified the instrument label across the manuscript as AAC-Q.

Revisions in Manuscript: Abstract (Results); Discussion (Section 4.1); Methods (Section 2.3.2). The definition text previously in Section 3.2 was deleted from Section 3.2 and incorporated into Section 2.3.2.

Comment 2. Reporting of multiplicity control for the UPI suicidal ideation item

Response.As requested, we changed from an α-level adjustment to a Bonferroni p-value adjustment for the family of planned correlations (number of correlations m = 16):

  • Methods (Section 2.4) (Data analysis) — added: “To address multiple comparisons for the planned correlation family (number of correlations m = 16), Bonferroni-adjusted p-values were computed as p_adj = p × 16 (values > 1 set to 1), and significance was evaluated at α = 0.05.”
  • Results (Section 3.2) (Correlations) — language unified to “after p-value adjustment” and added: “Bonferroni-adjusted p-values were calculated as p_adj = p × 16 (values > 1 set to 1), and statistical significance was evaluated at α = 0.05.”
  • Table 2 footnote — old “α = 0.0031” text deleted; added: “Bonferroni-adjusted p-values were computed (p_adj = p × 16; maximum 1), and significance was assessed at α = 0.05 for the planned 16 correlations involving AAC-Q.”
  • Wording cleanup. Phrases like “after correction” have been replaced with “after p-value adjustment” throughout the manuscript.
  • Re-computation. We recomputed adjusted p-values in SPSS; statistical conclusions were unchanged. Particularly, UPI suicidal ideation (r = 0.19), SCLD10 (r = 0.36), and perceived stress (r = 0.31) remained significant after p-value adjustment, whereas “difficulty in maintaining concentration” remained non-significant.

Revisions in Manuscript: Methods (Section 2.4); Results (Section 3.2); Table 2 footnote; wording unified to “after p-value adjustment” in Section 3.2 and Discussion (Section 4.2).

Round 3

Reviewer 3 Report

Comments and Suggestions for Authors

The authors have done a good job of responding to noted concerns. I am satisfied with the revisions and clarifications they have made. 

Author Response

We are very grateful to the reviewer for taking the time to re-evaluate our manuscript and for the encouraging feedback. We are pleased that the revisions and clarifications successfully addressed the earlier concerns. We greatly appreciate your recognition of our efforts, and we believe that your prior comments significantly improved the clarity, rigor, and readability of the manuscript.
